# EMBO *reports*

# Phosphorylation of myelin regulatory factor by PRKG2 mediates demyelination in Huntington's disease

Peng Yin[1,2,*] (iD), Qiong Liu[2,3], Yongcheng Pan[2,3], Weili Yang[1], Su Yang[1], Wenjie Wei[2,4], Xingxing Chen[2,5], Yan Hong[2], Dazhang Bai[1], Xiao-Jiang Li[1] & Shihua Li[1,**] (iD)

## Abstract

Demyelination is a common pathological feature of a large number of neurodegenerative diseases including multiple sclerosis and Huntington's disease (HD). Laquinimod (LAQ) has been found to have therapeutic effects on multiple sclerosis and HD. However, the mechanism underlying LAQ's therapeutic effects remains unknown. Using HD mice that selectively express mutant huntingtin in oligodendrocytes and show demyelination, we found that LAQ reduces the Ser259 phosphorylation on myelin regulatory factor (MYRF), an oligodendrocyte-specific transcription factor promoting the expression of myelin-associated genes. The reduced MYRF phosphorylation inhibits MYRF's binding to mutant huntingtin and increases the expression of myelin-associated genes. We also found that PRKG2, a cGMP-activated protein kinase subunit II, promotes the Ser259-MYRF phosphorylation and that knocking down PRKG2 increased myelin-associated protein's expression in HD mice. Our findings suggest that PRKG2-regulated phosphorylation of MYRF is involved in demyelination and can serve as a potential therapeutic target for reducing demyelination.

**Keywords** huntingtin; myelination; MYRF; oligodendrocytes; PRKG2
**Subject Categories** Molecular Biology of Disease; Neuroscience

## Introduction

Myelination, a process during which myelin wraps around axons, is critical for the development and function of the nervous system, as myelination vitally protects axon integrity and electrically insulates axons to enable its rapid conduction of action potential [1,2]. Myelin is constituted of abundant myelin-associated proteins such as myelin basic protein (MBP), myelin oligodendrocyte glycoprotein (MOG), and proteolipid protein (PLP), which are produced by oligodendrocytes [3]. In disease conditions, demyelination occurs and disrupts the nerve connection, leading to a variety of neurological diseases including multiple sclerosis [4,5]. In Huntington's disease (HD), the mutant huntingtin (HTT) protein carries an expanded polyglutamine (polyQ) repeat in its N-terminal region, which promotes aggregate formation in aged neuronal and glial cells and causes progressive neurodegeneration and neurological symptoms [6–8]. Growing evidence indicates that non-neuronal mutant HTT toxicity plays an important role in HD [6,8,9]. For example, defect in oligodendrocyte-enriched white matter is a typical characteristic feature in the early stages of HD patients [10–12].

Laquinimod (LAQ) was initially known as an immunomodulatory agent and was used to treat multiple sclerosis [13,14] and alleviate demyelination in the mouse models of different diseases [15–18]. LAQ has also been shown to improve behavioral phenotypes and white matter integrity in HD mice [19–21]. The neuroprotective effect of LAQ is well supported by its therapeutic effects on animal models of neuroinflammatory diseases, such as the experimental autoimmune encephalomyelitis [13,14], and is therefore thought to be due to an anti-inflammation effect [17,22]. However, the protective effect of LAQ on axons and myelination in HD mice was not found to associate with the anti-inflammatory effect [17,22]. Thus, the mechanism underlying the protective effects of LAQ remains unknown, and understanding this mechanism would be helpful for developing therapeutic strategies for brain diseases with demyelination.

We previously established a HD mouse model (PLP-150Q) that selectively expresses mutant HTT in oligodendrocytes and displays severe demyelination [23]. Thus, PLP-150Q mice allowed us to study the mechanism for demyelination. We found that LAQ could restore the expression of myelin-associated genes by reducing Ser259 phosphorylation in MYRF, an oligodendrocyte-specific transcription factor, and decreasing the subsequent binding of mutant HTT to MYRF. Increased Ser259 phosphorylation is accompanied by decreased

1 Ministry of Education CNS Regeneration Collaborative Joint Laboratory, Guangdong-Hongkong-Macau Institute of CNS Regeneration, Jinan University, Guangzhou, China
2 Department of Human Genetics, Emory University School of Medicine, Atlanta, GA, USA
3 Key Laboratory of Hunan Province in Neurodegenerative Disorders, Xiangya Hospital, Central South University, Changsha, China
4 Department of Oncology, Tongji Hospital, Tongji Medical College, Huazhong University of Science and Technology, Wuhan, China
5 Department of Physiology and Pathophysiology, Brain and Cognition Research Institute, Medical College, Wuhan University of Science and Technology, Wuhan, China
*Corresponding author. Tel: +86 2085 222157; E-mail: yinpeng177@163.com
**Corresponding author. Tel: +86 1371 6952480; E-mail: lishihualis@jnu.edu.cn

expression of MBP in the brain tissues of HD patients. We also found that cGMP-activated protein kinase subunit II (PRKG2) phosphorylates Ser259-MYRF and that knocking down PRKG2 increased MBP expression in PLP-150Q mouse. Our findings suggest that phosphorylation of MYRF mediates demyelination and can serve as a potential therapeutic target for reducing demyelination in neurodegenerative diseases.

## Results

### LAQ upregulated the myelin gene expression at the transcriptional level

We previously established transgenic mice that selectively express N-terminal HTT (1-212 aa) containing either 150Q (PLP-150Q) or 23Q (PLP-23Q) in oligodendrocytes under the control of the oligo-dendrocyte's specific proteolipid protein (PLP) promoter and found that PLP-150Q mice display robust demyelination [23]. Since LAQ can alleviate demyelination in the mouse models of different diseases [15–18] and since PLP-150Q mice express mutant HTT only in oligodendrocytes, we wanted to use PLP-150Q mice to investigate the specific effect of LAQ on mutant HTT-mediated demyelination. Three-month-old PLP-150Q and PLP-23Q mice were orally administrated with LAQ at two doses (5 or 25 mg/kg/day) or with the same volume of purified water (Vehicle) for 2 months (Fig EV1A). Consistent with the previous reports of the therapeutic effects of LAQ on different HD mouse models [19–21], treatment with LAQ (5 mg or 25 mg/kg/day) improved the rotarod and balance beam performance of PLP-150Q mice as compared with PLP-150Q mice treated with vehicle control (Fig EV1B), but did not alleviate the body weight reduction and early death of PLP-150Q mice (Fig EV1C). Because PLP-150Q mice only express mutant HTT in oligodendrocytes, the protective effects of LAQ indicate that LAQ could improve the function of oligodendrocytes to reduce neurological symptoms of HD mice and motivated us to further explore the mechanistic action of LAQ on oligodendrocytes.

Transmission electron microscopy revealed that a number of degenerated axons, which appeared swollen and dark, were present in PLP-150Q mice at the age of 5 months. LAQ (5 mg/kg) treatment

of 3-month-old PLP-150Q mice for 2 months, however, significantly improved the axon myelination (Fig 1A). Calculating g-ratios (the inner axonal diameter to the total outer diameter) verified that LAQ reduced this ratio or increased myelination of axons (vehicle: g = 0.7727 ± 0.0203 versus LAQ: g = 0.6418 ± 0.0191; *P < 0.05) (Fig 1B). We also crossed PLP-150Q mice to transgenic PLP-GFP mice and obtained PLP-150Q/PLP-GFP mice in which oligodendrocytes express both mutant HTT and GFP, allowing us to directly examine the integrity of GFP-labeled oligodendrocyte processes. We found that there was indeed reduced oligodendrocyte process length in PLP-150Q mice as compared with PLP-23Q mice, and LAQ (5 mg/kg) treatment for 2 months increased oligodendrocyte processes in their density and length in the corpus callosum of PLP-150Q mice (Fig EV1D). Interestingly, treatment with LAQ at 5 or 25 mg/kg could also restore the expression of myelin-related proteins in PLP-150Q mice, such as myelin basic protein (MBP), myelin-associated oligodendrocytic basic protein (MOBP), and myelin oligodendrocyte glycoprotein (MOG) (Fig 1C). However, in PLP-23Q mice, there were no significant changes of these myelin-associated proteins between the vehicle- and LAQ-treated groups (Fig EV2A), suggesting that LAQ selectively inhibited the toxicity of mutant HTT on myelin gene expression.

To explore whether myelin-associated genes are upregulated at the transcriptional level by LAQ, we used quantitative RT–PCR to examine the mRNA levels of myelin-associated genes. The results showed that, compared with PLP-23Q mouse, the levels of MBP and MOG transcripts were obviously decreased in PLP-150Q mice and were increased by LAQ (5 or 25 mg/kg) after treatment for 2 months (Figs 1D and EV2B). However, LAQ treatment did not alter the levels of aggregated and soluble huntingtin proteins when the antibodies to HTT (EM48) or polyglutamine repeats (1C2) were used to detect mutant HTT (Fig EV2C). These findings suggest that LAQ may antagonize the effect of soluble mutant HTT on suppressing the expression of myelin-associated genes.

### LAQ upregulated MBP transcription by dissociating MYRF from mutant HTT

We have previously found that mutant HTT abnormally binds myelin regulatory factor (MYRF), an oligodendrocyte-specific transcription

---

**Figure 1. LAQ increased the expression of myelin-associated proteins and reduced the binding of MYRF to mutant HTT in PLP-150Q mice.**

A   Electron microscopy revealed a number of demyelinated or degenerated axons in PLP-150Q mice at the age of 5 months. LAQ (5 mg/kg) treatment at 3 months of age for 2 months improved myelination in PLP-150Q mice. Scale bars: 2 μm (low magnification) and 0.5 μm (high magnification).

B   G-ratios, which were calculated and plotted against axon diameter with linear regression, were shown beneath the micrographs and were significantly decreased in PLP-150Q mice after LAQ treatment (G = 0.6418 ± 0.0191), compared with age-matched vehicle group (G = 0.7727 ± 0.0203). One-way ANOVA with Tukey's test. *P = 0.0166. At least 182 axons per genotype were examined from 3 mice in each group.

C   Western blotting showing that LAQ (5 or 25 mg/kg) treatment for 2 months upregulates multiple myelin proteins (MBP, MOBP, and MOG) in the corpus callosum in PLP-150Q mice, which were treated from 3 months of age. Ratios of MBP, MOBP, or MOG to vinculin obtained from 3 independent experiments were presented on the right. One-way ANOVA followed with Tukey's test. MBP: ***P = 0.0009; MOG: ***P = 0.0003; MOBP: **P = 0.0067. Data are mean ± SEM

D   Quantitative PCR of the transcript expression of myelin-associated genes (MBP and MOG) in PLP-150Q mouse corpus callosum at 5 months of age after LAQ (5 or 25 mg/kg) treatment for 2 months. Student's *t*-test. MBP: ***P = 0.0007; MOG: ***P = 0.0009. Data are mean ± SEM (n = 3).

E   The MBP DNA promoter was inserted into the pGL4.1 luciferase report vector and was co-expressed with N-terminal MYRF (nMYRF) and mHTT to assess its transcription activity via the luciferase assay. MYRF markedly enhanced the MBP promoter activity. N-terminal mutant HTT significantly inhibited the reporter activity, ***P < 0.001; which was reversed by LAQ (5 μM) treatment, **P < 0.01. The ratios were obtained from three independent experiments. One-way ANOVA followed with Tukey's test. Data are mean ± SEM.

F   Immunoprecipitation of transgenic mHTT from 3-month-old PLP-150Q mouse brains revealed that 5 or 25 mg/kg LAQ treatments for 2 months reduced the interaction between transgenic mutant HTT and MYRF. fMYRF: full-length MYRF; nMYRF: N-terminal MYRF. The ratio of immunoprecipitated MYRF to input obtained from three independent experiments was shown on the right. One-way ANOVA followed with Tukey's test. **P = 0.009. Data are mean ± SEM.

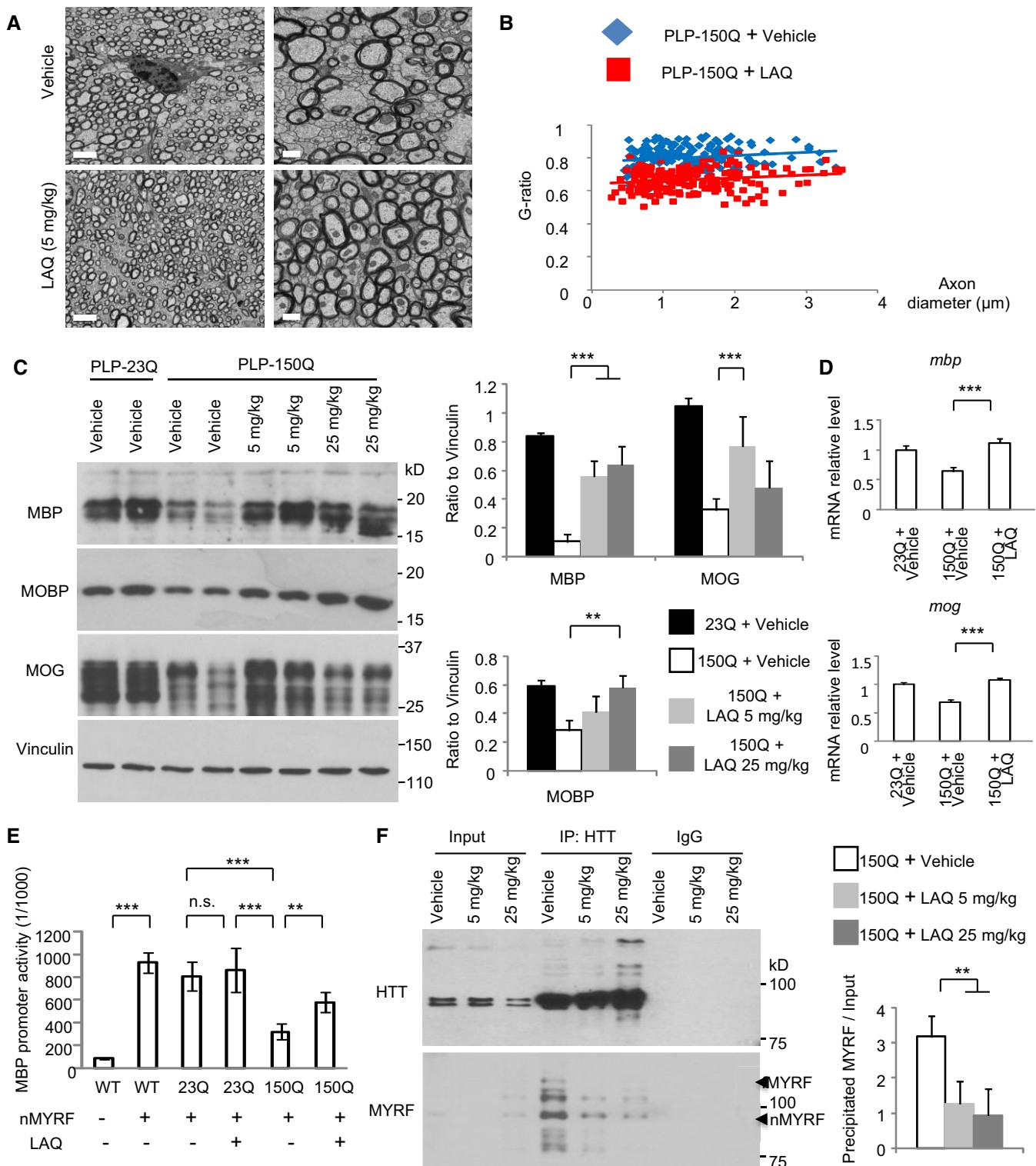

**Figure 1.**

factor, to affect its critical function for the regulation of transcription of myelin-associated genes [23]. It is known that N-terminal fragment of MYRF (nMYRF) is generated by cleavage of full-length MYRF in the cytoplasm and then moves into nucleus to activate myelin gene transcription in oligodendrocytes [24,25].

We then investigated whether LAQ influences the interaction of mutant HTT with MYRF to prevent its inhibitory effect on nMYRF. To address this issue, the MBP promoter was cloned into the reporter vector for a luciferase assay to examine its transcriptional activity on the basis of an earlier finding [26],

and this reporter was co-transfected with mutant HTT in HEK293 cells to assess the effect of mutant HTT. Although nMYRF co-transfection could significantly promote the reporter activity, co-expression of mutant HTT decreased this activity, and this decrease could be partially reversed by treating the transfected cells with 5 μM LAQ for 12 h (Fig 1E). We then performed *in vivo* immunoprecipitation of transgenic mutant HTT and MYRF from the corpus callosum of PLP-150Q mice. N-terminal MYRF was co-precipitated with mutant HTT; however, LAQ (5 or 25 mg/kg) treatment obviously attenuated this interaction (Fig 1F). The inhibitory effect of LAQ was further tested by using transfected HEK293 cells, which confirmed that 5 μM LAQ also diminished the binding of transfected mutant HTT (150Q-HTT) to MYRF (Fig EV2D).

### LAQ dephosphorylated MYRF to affect its interaction with mutant HTT

LAQ was found to regulate phosphorylation of various proteins [27,28], and transgenic expression of mutant HTT could lead to the increased phosphorylation of several proteins [29–32]. In PLP-150Q mice, we also found that expression of mutant HTT in oligodendrocytes resulted in the increased phosphor-NF-kB, -Akt or -JNK when compared with PLP-23Q mice and that 2-month treatment with LAQ (5 or 25 mg/kg) could significantly decrease the phosphorylation of these proteins in PLP-150Q mice (Fig 2A).

The above findings led us to investigate whether MYRF phosphorylation is involved in its interaction with mutant HTT such that LAQ may regulate its interaction via altering phosphorylation to alleviate the inhibitory effect of mutant HTT on myelin gene expression. We then performed *in vitro* phosphorylation assay by incubating GST-nMYRF with lysates from the oligodendrocyte-enriched corpus callosum. The results revealed that the lysates from the PLP-150Q mouse brain yielded strong phosphor-serine modification signals on GST-nMYRF as compared with the lysates

from the PLP-23Q mouse brain and that 5 mg/kg LAQ treatment eliminated this phosphorylation (Fig 2B, left panel). Furthermore, in the co-transfected HEK293 cells expressing nMYRF and mutant HTT, 5 μM LAQ treatment for 12 h or co-expression of the protein phosphatase, Λ-PPase, could attenuate the phosphorylation of the immunoprecipitated nMYRF and diminished its binding to mutant HTT (Fig 2C).

### Phosphorylation of Ser259 in MYRF is critical for its interaction with mutant HTT

There are multiple amino acids in MYRF for potential phosphorylation (Fig EV3A). We immunoprecipitated nMYRF with mutant HTT in HEK293 cells and isolated the nMYRF band (arrow in Fig 2D) for mass spectrometry to uncover its post-translational modifications (PTMs). The result identified a highly enriched phosphorylation signal on serine 259 (S259) in nMYRF (Fig EV3B). Replacing Ser259 with alanine (S259A) or its adjacent Ser261 with alanine (S261A) confirmed that only S259A substitution prevented the phosphorylation by the PLP-150Q mouse brain lysate (Fig 2B, right panel and Fig EV3C). We also transfected S259A or S261A nMYRF with mutant HTT (amino acid 1-212 HTT-150Q) in HEK293 cells to examine their transcriptional activity on the MBP promoter using luciferase assay and the interactions of different MYRF forms with mutant HTT. Indeed, nMYRF with S259A, but not S261A, substitution produced much higher transcriptional activity than wild-type nMYRF when mutant HTT was present (Fig 2E). Consistently, in co-transfected MYRF and mutant HTT cells, the immunoprecipitated S259A MYRF displayed very weak phosphorylation and a marked reduction in association with mutant HTT, while S261A MYRF remained to be phosphorylated and bound more mutant HTT (Fig 2F). All these findings suggest that S259 in MYRF can be phosphorylated and that this phosphorylation increases its interaction with mutant HTT and reduces its transcriptional activity.

---

**Figure 2. LAQ dephosphorylated N-terminal MYRF at pS259 to affect its interaction with mutant HTT.**

A  Western blotting of the phosphorylated NF-kB, NF-Akt, NF-JNK in PLP-150Q mouse corpus callosum at 3 months of age showing that LAQ (5 or 25 mg/kg) treatment for 2 months could dephosphorylate these proteins. Ratios of the phosphorylated proteins to their total proteins obtained from 3 independent experiments were presented on the right. One-way ANOVA with Tukey's test. NF-kB: ***P = 0.00067(left) and 0.00086(right); JNK: ***P = 0.00013(left) and 0.00047(right); Akt: **P = 0.0014 and ***P = 0.00012. Data are mean ± SEM.

B  *In vitro* phosphorylation assay of N-terminal MYRF (nMYRF). GST fusion proteins containing nMYRF were incubated overnight with brain lysates from PLP-23Q or PLP-150Q mice that were treated with vehicle or 5 mg/kg LAQ (left panel). GST fusion proteins containing wild-type N-terminal MYRF, S259A, or S261A were incubated with PLP-150Q mouse tissue lysates (right panel). The beads were then centrifuged and analyzed by Western blotting with anti-GST (lower panels) and anti-phosphor-serine (upper panels). Note that LAQ treatment could eliminate mutant HTT (PLP-150Q)-mediated MYRF phosphorylation and that S259A substitution prevents MYRF phosphorylation.

C  Co-immunoprecipitation of nMYRF and N-terminal HTT (1–212 aa) containing 150Q in HEK293 cells. Five μM LAQ treatment decreased the phosphorylation of immunoprecipitated MYRF and the amount of co-immunoprecipitated HTT. Λ-PPase served as the positive control for the dephosphorylation of MYRF. The ratios of the pho-serine MYRF or precipitated HTT to immunoprecipitated MYRF are shown under the blots, which were obtained from 3 independent Western blotting experiments. One-way ANOVA with Tukey's test. ***P = 6.51 × 10⁻⁵. Data are mean ± SEM.

D  Co-transfection of nMYRF with N-terminal mutant HTT in HEK293 cells and immunoprecipitation of nMYRF with or without LAQ (5 μM) treatment. Coomassie blue staining confirms the presence of immunoprecipitated N-terminal MYRF bands (arrow).

E  Transcriptional activity of wild-type N-terminal MYRF, S259A, or S261A with the MBP promoter reporter in HEK293 cells was detected using a luciferase assay. Compared to wild-type MYRF or S261A, S259A (non-phosphorylated) remained transcriptional activity that was not inhibited by N-terminal mutant HTT. The ratios were obtained from three independent experiments. One-way ANOVA with Tukey's test. ***P < 0.001. Data are mean ± SEM.

F  Co-transfection of wild-type N-terminal MYRF, S259A, or S261A with N-terminal mutant HTT in HEK293 cells and immunoprecipitation of MYRF. Compared to wild-type MYRF and S261A, less S259A (non-phosphorylated) was precipitated with mutant HTT. The ratios of the pho-serine MYRF or precipitated HTT to immunoprecipitated MYRF are shown under the blots and were obtained from three independent experiments. One-way ANOVA with Tukey's test. ***P = 2.30 × 10⁻⁵ Data are mean ± SEM.

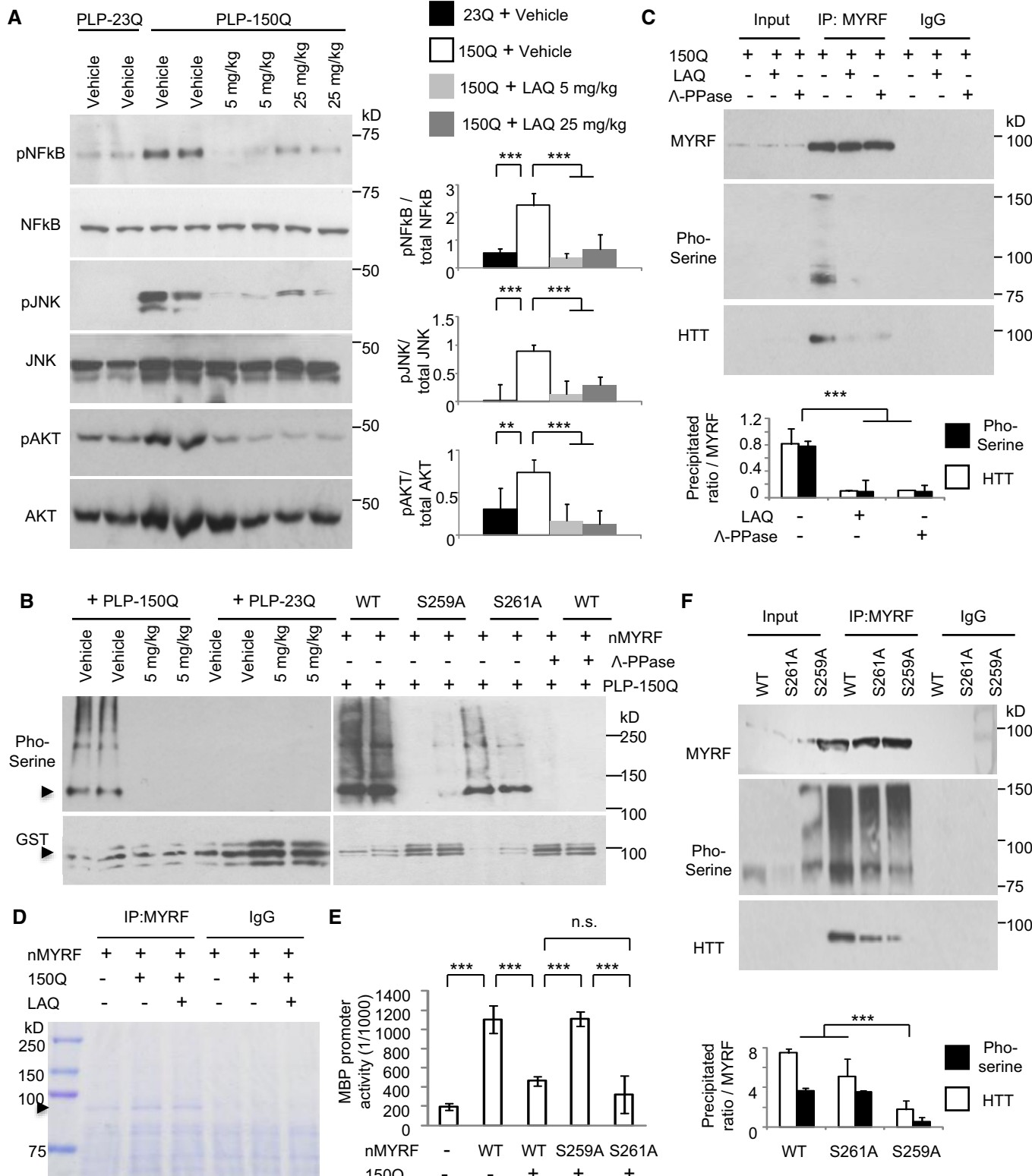

Figure 2.

**Increased MYRF phosphorylation in HD mouse and patient brains**

Next, we generated a specific antibody (anti-pS259) to phosphory-lated Ser259 in MYRF using a synthetic peptide containing amino

acids 256–269 (RKHpSESPPNTLNA) of mouse MYRF (Fig 3A). Western blotting demonstrated that anti-pS259 specifically recognized wild-type, but not S259A, nMYRF in transfected HEK293 cells (Fig EV3D). Furthermore, Western blotting (Fig 3B and C) and

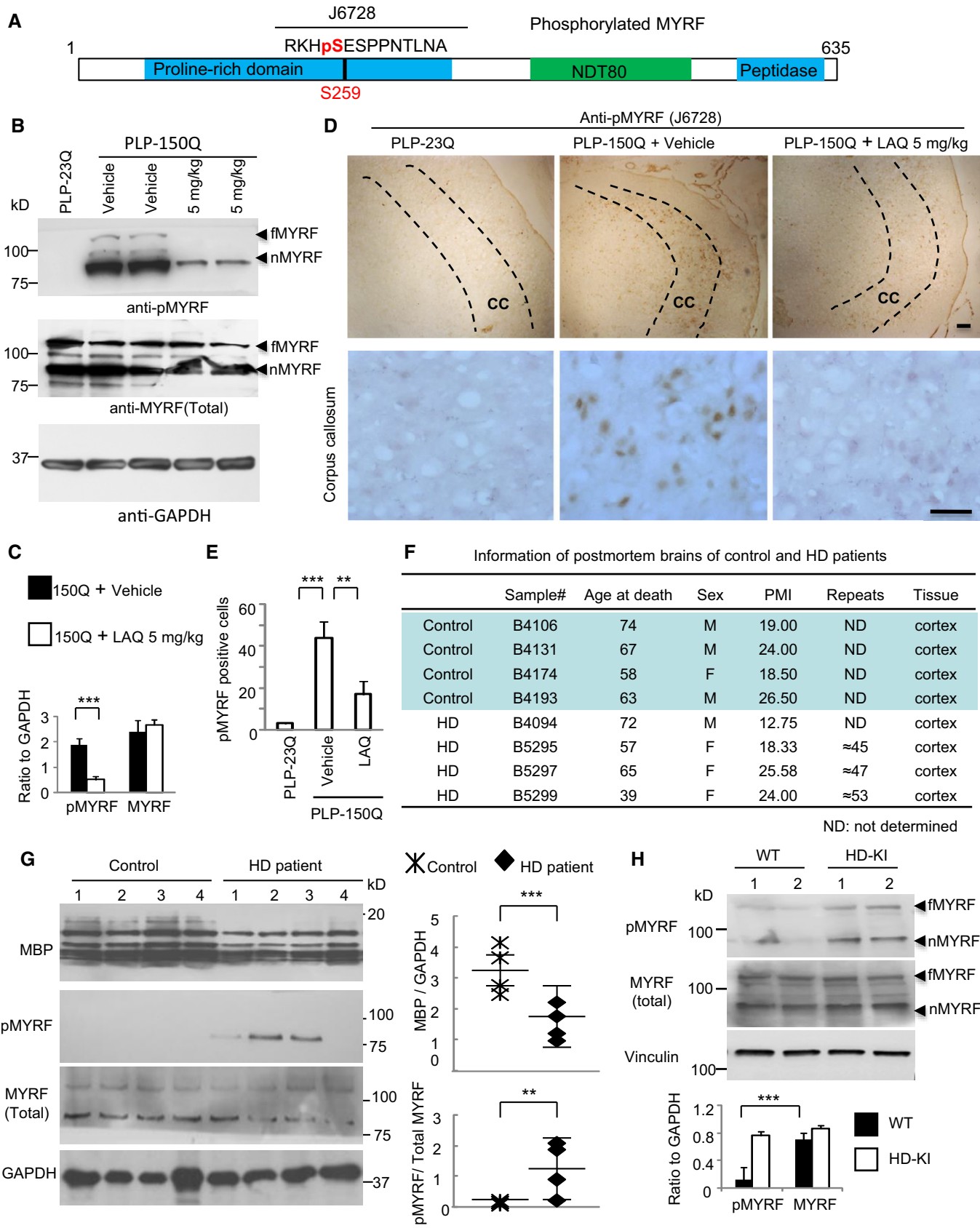

**Figure 3.**

◀

**Figure 3. Ser259-MYRF phosphorylation is increased in PLP-150Q mice and HD patient brains.**

A  Anti-Ser259-MYRF antibody was generated using the peptides containing pS259.
B  Western blotting analysis of the corpus callosum in PLP-23Q or PLP-150Q mice with anti-pS259 and an antibody to total MYRF. Note that LAQ (5 mg/kg) treatment for 2 months decreased the phosphorylation in both full-length (fMYRF) and N-terminal MYRF (nMYRF) in PLP-150Q mice.
C  Ratios of phosphor-S259-MYRF (pMYRF) and total MYRF to GAPDH. The ratios were obtained from three independent Western blotting experiments. Student's *t*-test; ***$P$ = 0.0007. Data are mean ± SEM.
D  Anti-Ser259-MYRF immunohistochemical staining of the corpus callosum (CC, indicated between two dotted lines) of PLP-23Q and PLP-150Q mice showing decreased expression of Ser259-MYRF by LAQ (5 mg/kg) treatment for 2 months. Micrographs at 10× (upper) and 40× (lower) are shown. Scale bars: 40 μm.
E  Quantitative statistics of the cells stained by anti-Ser259 MYRF in each field was presented. Twenty random fields (40×) of each section were examined to count anti-Ser259-positive cells, *n* = 3 mice in each group. One-way ANOVA with Tukey's test. ***$P$ = 0.0003; **$P$ = 0.0039. Data are mean ± SEM.
F  Information of post-mortem brains of control and HD patients.
G  Western blotting analysis of MBP and Ser259-MYRF expression in the post-mortem brain cortex tissues in 4 HD patients and 4 non-HD control individuals. Ratios of MBP to GAPDH, or Ser259 (pMYRF) to total MYRF are presented on the right. One-way ANOVA with Tukey's test. MBP: ***$P$ = 0.0004; pMYRF: **$P$ = 0.0051. Data are mean ± SEM.
H  Western blotting analysis of the expression of Ser259 (pMYRF) and total MYRF (full-length: fMYRF and N-terminal: nMYRF) in 13-month-old HD-KI and WT mouse cortex. Ratios of pMYRF to total MYRF obtained from three independent experiments were presented beneath the blots. One-way ANOVA with Tukey's test. ***$P$ = 0.00083. Data are mean ± SEM.

immunohistochemistry (Fig 3D and E) demonstrated that anti-pS259 could also recognize phosphorylated MYRF in the PLP-150Q mouse corpus callosum tissue, and this phosphorylated signal was inhibited by 5 mg/kg LAQ treatment. However, total MYRF did not display obvious difference in the PLP-150Q mouse corpus callosum before and after LAQ treatment, suggesting that LAQ only alters MYRF phosphorylation (Fig EV3E).

The increased Ser259-MYRF phosphorylation in the PLP-150Q mouse brain led us to examine whether this increase also occurs in the post-mortem brain tissues from HD patients that were obtained from Harvard Brain Tissue Resource Center (HBTRC) (Fig 3F). Western blotting revealed that 3 of 4 brain tissues from HD individuals displayed clear Ser259-MYRF signals while none of control samples showed detectable phosphorylated MYRF (Fig 3G). In addition, MBP expression was also noticeably decreased in the brain tissues of patients with HD (Fig 3G). The results of HD patient brains support the observation that MYRF phosphorylation is increased while expression of myelin-associated genes is decreased in the PLP-150Q mouse brain.

We previously reported that HD knock-in (KI) mice, which express full-length mutant HTT with 140Q, showed obvious axonal degeneration as an early pathologic event [33]. To investigate whether MYRF phosphorylation is also increased in HD-KI mice, we examined the tissues from the corpus callosum from two 13-month-old HD-KI mice and two age-matched WT mice via Western blotting. The results also demonstrated the increased Ser259-MYRF phosphorylation in HD-KI mice (Fig 3H).

### PRKG2 is the kinase to mediate the S259-MYRF phosphorylation

Since LAQ has not been reported to directly modulate protein phosphorylation, we assumed that it may alter gene expression to influence protein phosphorylation, based on the previous finding that LAQ modulates gene transcription in the mouse brain [14]. We therefore performed microarray analysis of gene expression from the oligodendrocyte-enriched corpus callosum of 3-month-old PLP-150Q mice that had been treated with 5 mg/kg LAQ for 2 months. Results revealed that 27 genes were significantly upregulated and 103 genes significantly downregulated after LAQ treatment as compared with the vehicle group (Fig 4A). The potentially altered genes were categorized based on their functions (Fig 4B), and the

upregulated genes that are involved in protein phosphorylation were analyzed further (Fig 4C).

Using three predictive programs (NetPhos3, KinasePhos, and GPS3) to identify the most prominent candidate for phosphorylation of Ser259-MYRF, we identified PRKG2 (cGMP-dependent protein kinase type II, cGKII) as the top candidate for Ser259-MYRF phosphorylation (Fig 4D). The PRKG2 substrate consensus core sequences $(R/K)_{2-3}$-X-S/T-X, where R = arginine, K = lysine, T = threonine, S = serine, X standing for any random amino acid) are mostly consistent with Ser259-MYRF's surrounding peptide sequences "KKRKHSE" (amino acid 254-260), which are well conserved in mouse, rat, monkey, and human (Fig 4D).

### LAQ suppresses PRKG2 gene transcript expression via aryl hydrocarbon receptor (AhR)

Next, we wanted to examine whether PRKG2 is responsible for Ser259-MYRF phosphorylation and that LAQ may affect its expression in PLP-150Q mice. We found that, compared with PLP-23Q mice, the protein level of both total and active forms of PRKG2 was increased in the corpus callosum of PLP-150Q mice (Fig 4E). Consistently, the phosphorylation of a known PRKG2 substrate, the vasodilator-stimulated phosphoprotein (VASP), was also increased in PLP-150Q mice (Fig 4E). In cultured N2a mouse cell line, we also verified that the transfection of mutant HTT (150Q-HTT) promoted PRKG2's expression and its kinase activity as compared with 23Q-HTT (Fig EV4A). Moreover, qRT–PCR showed that 5 mg/kg LAQ inhibited the transcription of PRKG2 (Fig 4F) and reduced the expression of PRKG2, which is evidenced by a 1.66-fold down-regulation in the microarray results. Western blotting also showed that LAQ reduced PRKG2 kinase activity, leading to a reduced phosphorylation of VASP (Fig 4E). Thus, it is likely that PRKG2 is upregulated in oligodendrocytes that express mutant HTT but this activation can be suppressed by LAQ via inhibiting the transcription of PRKG2 mRNA.

Since down-regulation of PRKG2 gene by LAQ is at the transcriptional level, the potential DNA promoter regions of PRKG2 were isolated for luciferase assays in transfected N2a cells. We found that the pPRKG2 promoter region (−472/+131) had the highest reporter activity (Fig 5A), which could be increased by mutant HTT's (150Q-HTT) transfection (Fig EV4B) but inhibited by LAQ (5 or 10 μM) (Fig 5B). By searching for the potential transcription factors

that can bind to the PRKG2 promoter, six binding sites ("CACGC" or "GCGTG") for the aryl hydrocarbon receptor (AhR) were identified (Fig 5C). AhR is a transcriptional factor that can either inhibit gene expression [34–36] or increase Cyp1a1 and Ugt1a6a [37] gene

transcription, and its binding to DNA is influenced by LAQ [14,21]. Using qPCR, we confirmed the activation of AhR and its downstream molecules Cyp1a1 and Ugt1a6a by LAQ treatment (Fig 5D). Furthermore, knocking down AhR in N2a cells using siRNA

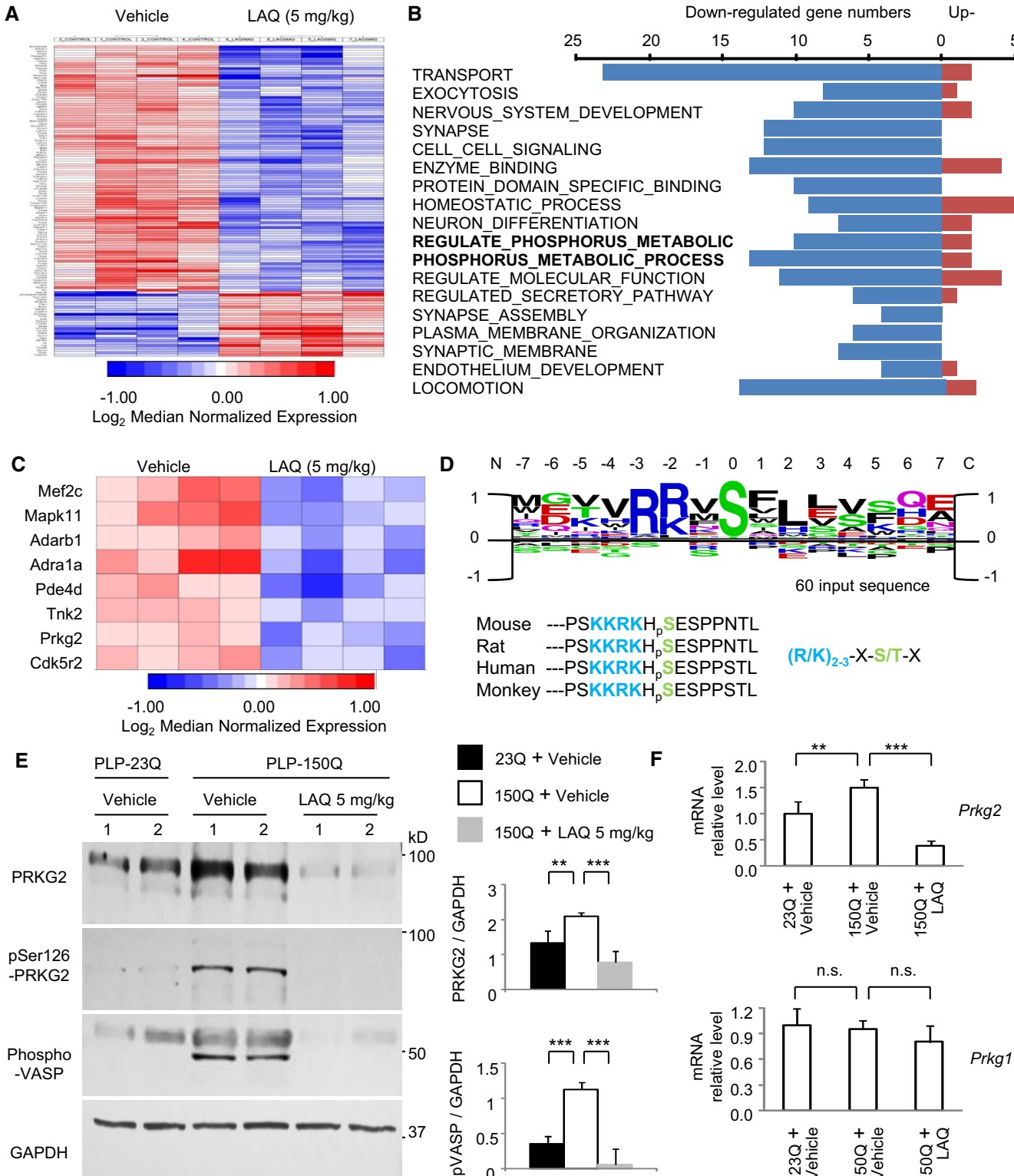

**Figure 4.**

**Figure 4.  PRKG2 is the potential candidate kinase for phosphorylation of S259 in MYRF.**

A  Heatmaps of gene expression in PLP-150Q mouse corpus callosum at 5 months after LAQ (5 mg/kg) treatment for 2 months revealed 27 significantly upregulated and 103 significantly downregulated genes.

B  Most significant gene networks of different functions via GO-cluster and categorization. The bold-type indicated two phosphorylation associated clusters of "REGULATE_OF_PHOSPHORUS_METABOLIC_PROCESS" and "POSITIVE_REGULATE_OF_PHOSPHORUS_METABOLIC_PROCESS".

C  The individual networks of upregulated gene were heat-mapped typically, on the clusters of "REGULATE_OF_PHOSPHORUS_METABOLIC_PROCESS" and "POSITIVE_REGULATE_OF_PHOSPHORUS_METABOLIC _PROCESS".

D  Identification of consensus sequences from PRKG2 kinase substrate peaks, in which the same motif was identified using the 15 amino acid sequences surrounding the strongest 60 peaks. The PRKG2 substrate core consensus sequences of $(R/K)_{2-3}$-X-S/T-X, where S = serine, R = arginine, K = lysine, T = threonine, X standing for any random amino acid, are mostly consistent with MYRF amino acid 252-266. The rat, monkey, and human MYRF have the highly conserved core sequences of "KKRKHSE".

E  Western blotting showing down-regulation of PRKG2 in PLP-150Q mouse corpus callosum at 3 months of age by 5 mg/kg LAQ. Ratios of PRKG2 or phosphorylated VASP (p-VASP) to GAPDH obtained from three independent experiments were presented on the right. One-way ANOVA with Tukey's test. PRKG2: $**P = 0.0076$ (left) and $***P = 0.0015$ (right); p-VASP: $***P = 3.86 \times 10^{-5}$ (left) and $1.01 \times 10^{-5}$ (right). Data are mean $\pm$ SEM.

F  Quantitative PCR of PRKG2 or PRKG1 gene expression in PLP-150Q mouse corpus callosum at 5 months of age after LAQ (5 mg/kg) treatment for 2 months. Only PRKG2 was decreased by LAQ. $n = 3$ mice in each group. Student's $t$-test; $**P = 0.0073$; $***P = 0.00041$. Data are mean $\pm$ SEM.

interference could block the inhibitory effect of LAQ on the PRKG2 promoter activity (Figs 5E and EV4C). Using chromatin immunoprecipitation (ChIP) assay, we immunoprecipitated AhR-associated DNAs in the presence of 10 μM LAQ or 10 nM 2,3,7,8-tetrachlorodibenzo-p-dioxin (TCDD), a known AhR agonist, and measured the associated PRKG2 promoter DNAs. We found that LAQ significantly promoted the association of AhR with the PRKG2 promoter (Fig 5F). All these results suggest that LAQ suppresses PRKG2 expression via AhR's negative regulation.

**PRKG2 phosphorylates Ser259 in MYRF**

To test the idea that PRKG2 mediates the phosphorylation of pS259 in MYRF, we first performed *in vitro* assay using cultured N2a cells and found that MYRF was obviously phosphorylated by the PKG activator, 8-Br-cGMP (10 or 100 μM), but not by the PKG inhibitors, RKRARKE or KT5823 (Fig 6A). Importantly, co-transfection with mutant HTT also increased MYRF phosphorylation, which could be markedly inhibited by RKRARKE or KT5823 (Fig 6A). This *in vitro* finding, which suggests that expression of mutant HTT may directly or indirectly activate PRKG2 to phosphorylate MYRF, is consistent with the *in vivo* evidence of the elevated PRKG2 (Fig 4E) and MYRF phosphorylation (Fig 3G) in the HD brains.

Next, we generated an AAV vector expressing full-length mouse PRKG2 cDNA under the control of the CMV promoter and confirmed its expression in cultured N2a cells (Fig 6B). Overexpression of AAV-PRKG2 indeed significantly promoted the phosphorylation of MYRF in cultured N2a cells (Fig 6C). We then stereotaxically injected AAV-PRKG2 or control AAV-GFP into the corpus callosum of 2-month-old WT mice (Fig 6D). As expected, immunohistochemical (Figs 6E and F, and EV5A and B) and immunofluorescent (Fig 6G) staining showed that AAV-PRKG2 expression promoted Ser259-MYRF phosphorylation whereas AAV-GFP did not. Furthermore, Western blotting with the antibody (pS295-MYRF) to Ser259-MYRF validated that AAV-PRKG2, but not AAV-GFP, could lead to the phosphorylation of Ser259-MYRF in the mouse brain (Fig 6H).

**Knocking down PRKG2 alleviated demyelination and reduced Ser259-MYRF phosphorylation in PLP-150Q/Cas9 mouse**

If Ser259 phosphorylation in MYRF is mediated by PRKG2 and inhibition of PRKG2 expression by LAQ contributes to the alleviation of demyelination in PLP-150Q mice, knocking down PRKG2 should also lead to the similar outcome. To silence the endogenous PRKG2 in PLP-150Q mouse brain, we used CRISPR/Cas9 to deplete the

**Figure 5.  LAQ suppresses PRKG2 gene transcript expression via AhR negative regulation.**

A  The different PRKG2 promoter regions were inserted into the pGL4.1 luciferase report vector. The core promoter reporter vector of PRKG2 (−472/+131) was determined for transcription activity in cultured N2a cells, obtained from three independent experiments. One-way ANOVA with Tukey's test; $***P < 0.001$. Data are mean $\pm$ SEM.

B  The treatment of LAQ (5 or 10 μM) caused a marked inhibition on the transcription activity of the PRKG2 promoter detected by the luciferase assay from three independent experiments. One-way ANOVA with Tukey's test. $***P < 0.001$. Data are mean $\pm$ SEM.

C  The putative cis-elements (in box and sequences below) in the PRKG2 promoter were determined by TFSEARCH, ConTra, and ALGGEN program analyses. Six AhR transcriptional factor binding sites were predicted according to the conserved sequence of "CACGC" or "GCGTG".

D  The qPCR analysis of the transcripts of Cyp1a1 and Ugt1a6a, which were mediated by AhR, in the PLP-150Q mouse brain (corpus callosum). Note that Cyp1a1 and Ugt1a6a were significantly increased by LAQ (5 mg/kg). $n = 3$ mice in each group. Student's $t$-test; Cyp1a1: $***P = 0.00042$; Ugt1a6a; $***P = 0.00057$. Data are mean $\pm$ SEM.

E  The PRKG2 core promoter activity in N2a cells transfected with AhR siRNA or its scrambled siRNA control and then treated with 5 μM LAQ. The values of promoter activity via luciferase report assay were obtained from three independent experiments. One-way ANOVA with Tukey's test. $***P < 0.001$. Data are mean $\pm$ SEM.

F  The semi-PCR detection of PRKG2 promoter DNAs associated with AhR that was immunoprecipitated by anti-AhR in ChIP assay. The N2a cells were treated with 10 μM LAQ or 10 nM 2,3,7,8-TCDD or DMSO for 12 h. The quantification of PRKG2 promoter DNAs associated with AhR that was immunoprecipitated by anti-AhR in ChIP assay. The N2a cells were treated with 10 μM LAQ or 10 nM 2,3,7,8-TCDD or DMSO for 12 h. The results were obtained from three independent experiments. One-way ANOVA with Tukey's test. LAQ: $***P = 0.00065$; 2,3,7,8-TCDD: $***P = 0.00014$. Data are mean $\pm$ SEM.

Source data are available online for this figure.

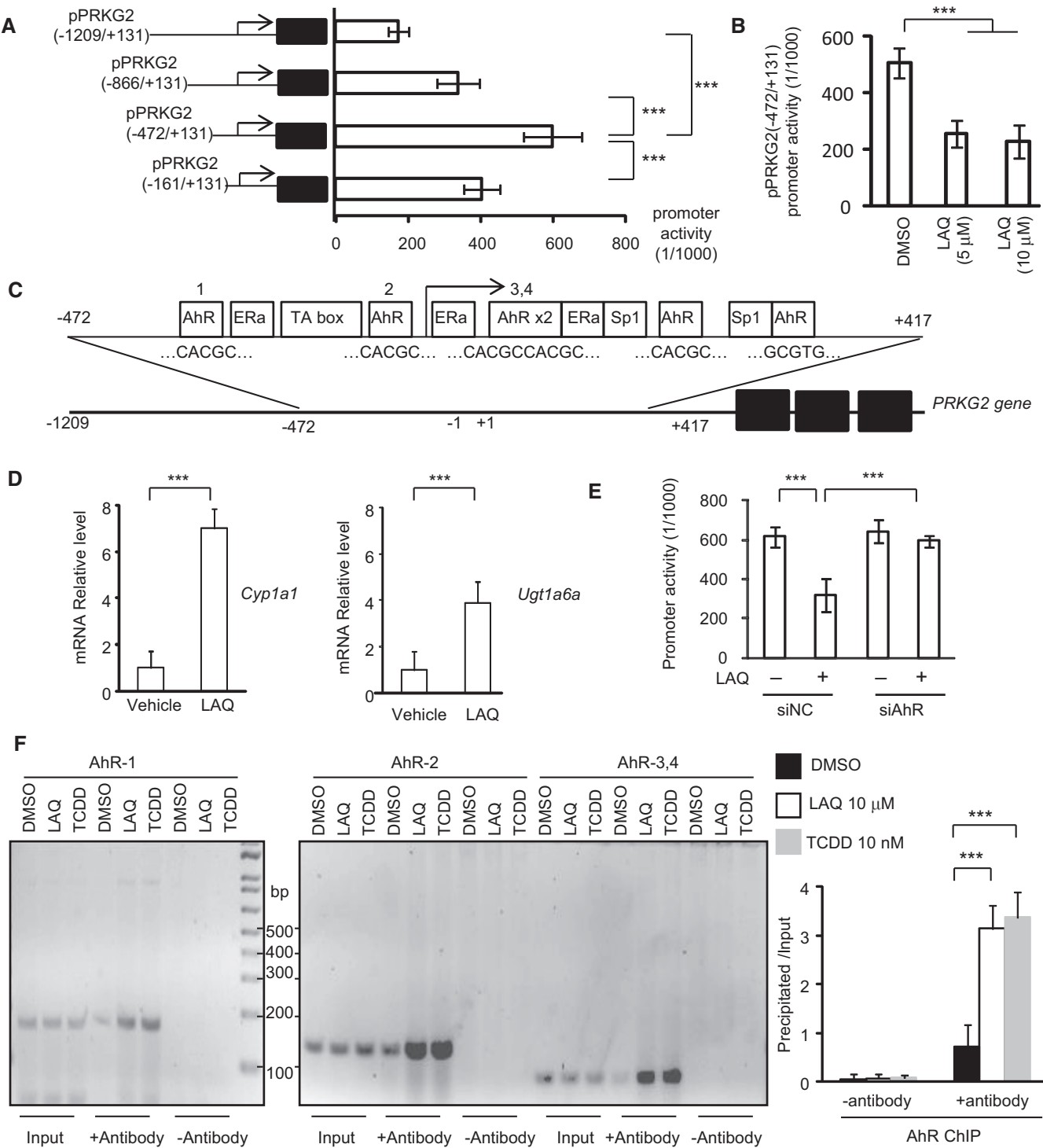

**Figure 5.**

expression of PRKG2 by stereotaxically delivering AAV-PRKG2-gRNA into the brains of PLP-150Q/Cas9 mice, which were generated by crossing PLP-150Q mice with transgenic Cas9 mice [38]. The expression of Cas9 and mutant HTT was confirmed by Western blotting of the injected brain tissues in PLP-150Q/Cas9 mice (Fig EV5C). The gRNAs for PRKG2 were designed to target exon 1 of the mouse PRKG2 gene and were expressed in AAV-U6-gRNA-CMV-RFP vector

(Fig 7A). Genome-editing activity of PRKG2 gRNA was confirmed by T7E1 assay (Fig EV5D) and sequencing of PCR products (Fig EV5E). AAV-PRKG2-gRNA virus was then injected into the corpus callosum of PLP-150Q/Cas9 mice (Fig 7A). Western blotting (Fig 7B) and immunohistochemistry (Fig 7C and D) showed that injection of AAV-PRKG2-gRNA decreased the protein level of PRKG2 and inhibited the phosphorylation of Ser259-MYRF in the mouse

brain. More importantly, knocking down PRKG2 increased the expression of MBP (Fig 7B). The increased MBP was further verified by immunofluorescent staining of the corpus callosum in PLP-150Q/Cas9 mice when compared with control injection with AAV-gRNA (Fig 7E). Quantification of the relative levels of fluorescent signals also confirmed the increase in MBP expression (Fig 7F). Based on these findings, we propose that LAQ reduces the expression of PRKG2 via AhR-related negative regulation, therefore

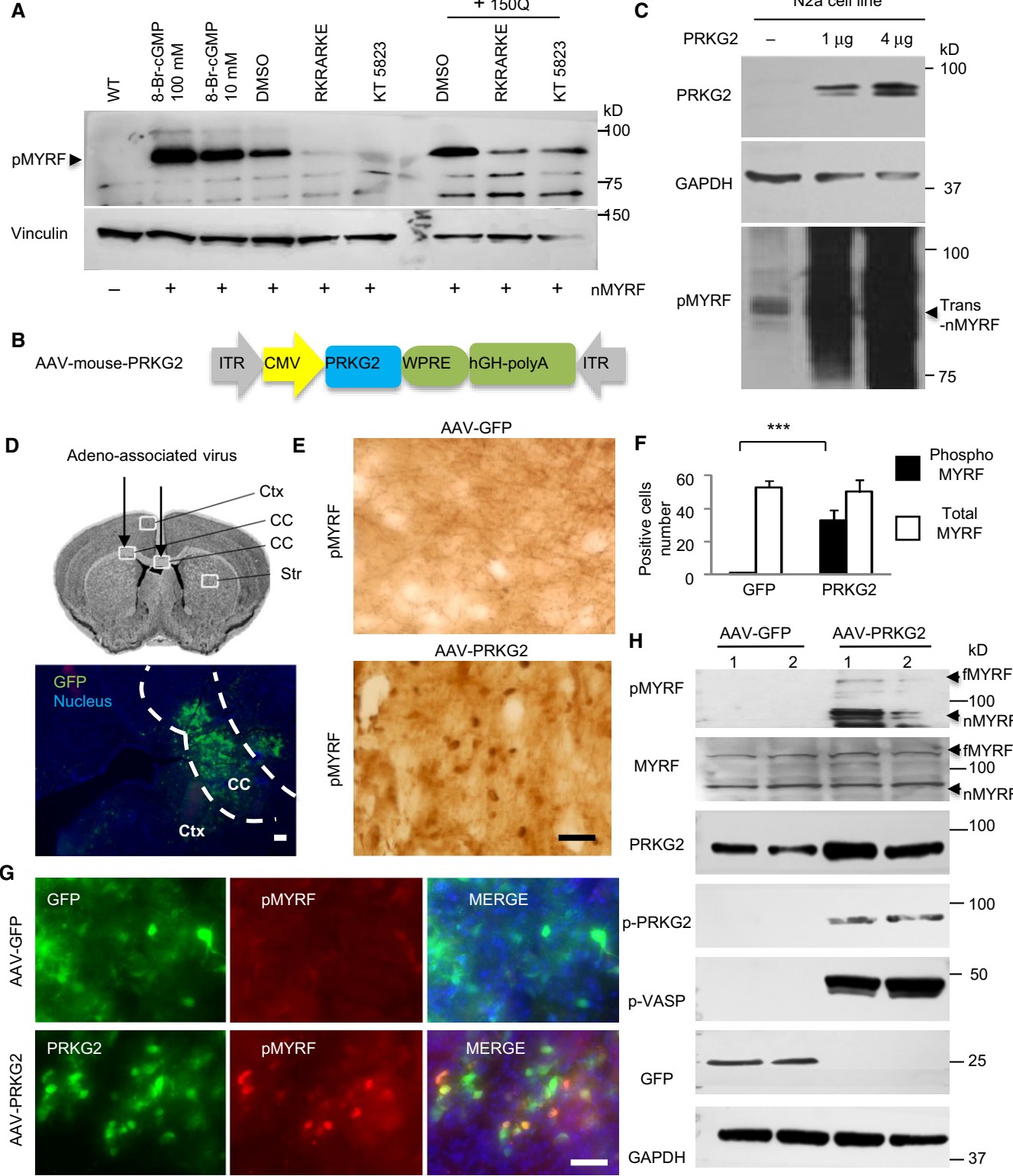

**Figure 6.**

◀

**Figure 6.  The activation or overexpressing of PRKG2 promoted the phosphorylation of Ser259-MYRF.**

A  Western blotting showing that the phosphorylation of transfected N-terminal MYRF (pMYRF) was promoted by treatment with the PKG activator of 8-Br-cGMP (10 or 100 μM) for 1 h or co-transfected 150Q-HTT for 24 h, which was blocked by treatment of the PKG inhibitors of RKRARKE (50 μM) or KT5823 (5 μM) for 12 h. trans-nMYRF: transfected N-terminal MYRF.

B  Adeno-associated virus vector expressing full-length PRKG2 under the control of the CMV promoter.

C  Western blotting of N2a cell line that was transfected with AAV-PRKG2 for 48 h. The blots were probed with anti-Ser259 (pMYRF) antibody. Note that overexpression of PRKG2 increased phosphorylation of N-terminal MYRF (nMYRF) in a dose-dependent manner.

D  Stereotaxic injection of AAV-GFP virus into the corpus callosum (CC, indicated between two dotted lines) in the mouse brain. Ctx: cortex, Str: striatum. Scale bar: 100 μm.

E  Immunohistochemical staining with anti-Ser259 (pMYRF) showing that AAV-PRKG2, but not AAV-GFP, promotes MYRF phosphorylation in the corpus callosum. Scale bar: 40 μm.

F  Quantitative analysis of the pMYRF- or total MYRF-positive cells in each field (40×). Twenty random fields in each section were examined, $n = 3$ mice in each group. One-way ANOVA with Tukey's test; $***P = 8.79 \times 10^{-5}$. Data are mean ± SEM.

G  Double immunofluorescent staining with antibodies to PRKG2 and Ser259 showing that AAV-PRKG2, but not AAV-GFP (green), promotes MYRF phosphorylation (red) in the injected corpus callosum. Scale bar: 40 μm.

H  Western blotting analysis of the mouse corpus callosum tissues using anti-Ser259 (pMYRF), phosphor-PRKG2 (pPRKG2), or phosphor-VASP (pVASP). Note that overexpression of mouse PRKG2 leads to the phosphorylation of MYRF and VASP. fMYRF: full-length MYRF, nMYRF: N-terminal MYRF.

Source data are available online for this figure.

resulting in decreased MYRF phosphorylation and its dissociation from mutant HTT to restore the normal function of the MYRF in oligodendrocytes and also to activate the transcript expression of myelin-associated genes (Fig 7G).

## Discussion

Demyelination or white matter abnormality is an important pathological feature in the brains of patients with HD in the early or pre-symptomatic stages [10–12]. Our previous studies have demonstrated that mutant HTT abnormally binds MYRF in oligodendrocytes to affect the expression of myelin-associated genes [23]. The new findings in the current study are that this binding is dependent on phosphorylation of Ser259-MYRF and that LAQ can reduce this phosphorylation to alleviate the toxic effect of mutant HTT on MYRF-mediated myelin gene transcription. Moreover, we identified PRKG2 as a kinase that phosphorylates Ser259-MYRF to promote its binding to mutant HTT and to inhibit the transcriptional activity of MYRF on myelin protein expression.

LAQ is an oral drug that can diffuse freely across the blood–brain barrier and executes its protective effects in multiple sclerosis [39]. Since LAQ has broad actions for its immunomodulatory, anti-inflammatory, and neuroprotective effects [39], the mechanism of its protective effects on demyelination has not been defined. In the current studies, we examined the effect of LAQ in PLP-150Q mice that selectively express mutant HTT in oligodendrocytes. We also compared PLP-150Q mice with PLP-23Q mice that express normal HTT in oligodendrocytes. Thus, the alleviation of the phenotypes of PLP-150Q mice by LAQ should be specifically related to the effect of LAQ on mutant HTT in oligodendrocytes. In addition, we focused on MYRF and MYRF-mediated expression of myelin genes such as MBP. Since MYRF and MBP are only expressed in oligodendrocytes and since the expression of mutant HTT in PLP-150Q mice is restricted to oligodendrocytes, we could identify the selective effects of mutant HTT and LAQ treatment in oligodendrocytes by comparing with PLP-23Q mice that express normal N-terminal HTT in oligodendrocytes.

Our studies revealed that LAQ treatment resulted in the improvement of myelination of PLP-150Q mice and their motor function as well as increased myelin gene expression. However, the rescuing effect of LAQ is partial and did not completely prevent the body weight reduction and early death of PLP-150Q mice. The results suggest that although oligodendrocytes play a critical role in myelination, the orchestration of brain myelination is complex and multifactorial, as this process is regulated by a number of distinct factors such as extracellular growth factors, matrix proteins, and electrical activity [40,41]. It is also known that oligodendrocytes release neurotrophic factors, neurotransmitters, and immunomodulatory molecules [42,43]. Given that mutant HTT can interact with a variety of proteins to affect multiple intracellular functions [44,45], it is possible that LAQ-mediated increase in the expression of myelin-associated genes can only partially rescue phenotypes while other cellular dysfunctions remain in mutant HTT-expressing oligodendrocytes.

The important finding in the current study is that MYRF is phosphorylated at S259 by PRKG2. This phosphorylation appears to be related to demyelination phenotypes. First, the phosphorylation of MYRF promotes its binding to mutant HTT and reduces its activation on the MBP DNA promoter. As a result, the MYRF phosphorylation is accompanied by the decreased expression of MBP in the brains of PLP-150Q mice. Second, the phosphorylation of Ser259-MYRF is increased in the HD mouse brains due to the increased PRKG2 expression and activity. Although the mechanism behind this increase remains to be investigated, this increase may reflect the toxic effect of accumulation of misfolded proteins in oligodendrocytes. It is possible that misfolded protein accumulation in oligodendrocytes increases ER stress to alter phosphorylation of a number of proteins as reported in other types of cells [30,31,46]. Increased phosphorylated PRKG2 seen in the PLP-150Q brain may account for Ser259-MYRF phosphorylation. Third, altered expression of PRKG2 can also influence MYRF phosphorylation in cultured cells and in the mouse brain. Reducing PRKG2 could inhibit Ser259-MYRF phosphorylation and increase MBP expression in the PLP-150Q mouse brain, suggesting that PRKG2 is responsible for the Ser259-MYRF phosphorylation.

PRKG2 activates nitric oxide (NO)/cGMP/PKG cascade [47,48], whose activation plays a deteriorative role in the pathophysiology of multiple sclerosis [49,50]. Increased NO activity and cGMP level

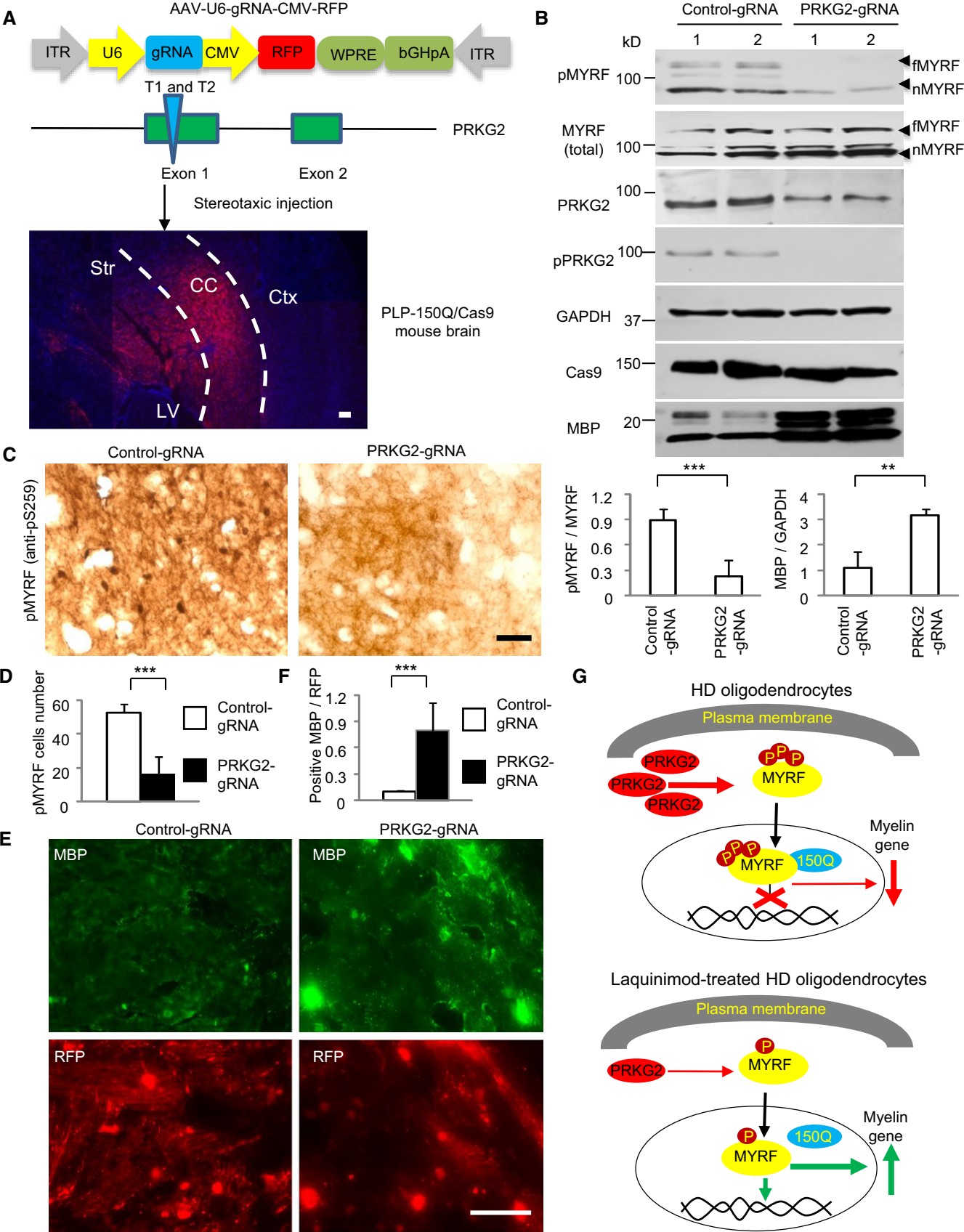

**Figure 7.**

**Figure 7. Knocking down PRKG2 alleviated demyelination and decreased phosphorylation of MYRF in PLP-150Q/Cas9 mice.**

A  Construction of AAV-U6-gRNA-CMV-RFP vector expressing PRKG2 gRNA under the U6 promoter and RFP protein under the CMV promoter, respectively. AAV-gRNA-PRKG2 was injected into the corpus callosum (CC, indicated between two dotted lines) in PLP-150Q/Cas9 mouse at 2 months of age and the injected tissues were isolated 4 weeks after injection (lower panel). Ctx: cortex, Str: striatum, LV: lateral ventricle. Scale bar: 100 μm.

B  Western blotting analysis of the PLP-150Q/Cas9 mouse corpus callosum tissues using the antibody to Ser259 (pMYRF), phosphor-PRKG2, or MBP. Note that reduction of mouse PRKG2 by CRISPR/Cas9 leads to the decreased MYRF phosphorylation and the increased expression of MBP. The ratios of pMYRF to MYRF or MBP to GAPDH obtained from three independent experiments were presented under the blots. Student's $t$-test; pMYRF: ***$P = 0.00026$; MBP; **$P = 0.0036$. Data are presented as mean ± SEM. fMYRF: full-length MYRF, nMYRF: N-terminal MYRF.

C  Immunohistochemical staining with antibody to Ser259 showed that knocking down PRKG2 reduced MYRF phosphorylation in the corpus callosum of PLP-150Q mouse. Scale bar: 40 μm.

D  Quantitative analysis of the pMYRF-positive cells in each field (40×). Twenty random fields in each section were examined, $n = 3$ mice in each group. One-way ANOVA with Tukey's test; ***$P = 0.00022$. Data are mean ± SEM.

E  Immunofluorescent staining with antibodies to Ser259 or MBP showed the AAV- PRKG2-gRNA (red) injection increased MBP expression (green) as compared with AAV-gRNA-control. Scale bar: 40 μm.

F  Quantitative analysis of the MBP-positive cells in the corpus callosum injected with control-gRNA or PRKG2-gRNA. Twenty random fields (40×) in each section were examined. $n = 3$ mice in each group. One-way ANOVA with Tukey's test; ***$P = 4.43 \times 10^{-5}$. Data are mean ± SEM.

G  A proposed model for the protective effect of LAQ in HD mice. LAQ increases myelin protein expression by reducing the expression of PRKG2, leading to reduced phosphorylation of MYRF and its dissociation from mutant HTT and therefore increasing its transcriptional activity to express myelin-associated genes.

Source data are available online for this figure.

were evident in the serum of MS patients [51–53]. Activated microglia can secret NO to affect oligodendrocytes [54,55], and MS lesions are characterized by the presence of activated microglia and their secretion of NO, which contribute to chronic neuroinflammation [56]. LAQ may reduce PRKG2 to yield inhibitory effects on glial activation for treating MS. Our findings open up a new avenue to further investigate the role of PRKG2 in MS and other neurodegenerative diseases. Identification of the increased Ser259-MYRF phosphorylation in the HD mouse and patient brains will also lead to the investigation of whether this increased Ser259-MYRF phosphorylation occurs in other diseases with the similar demyelination feature. Thus, our findings suggest that PRKG2 and the phosphorylation of Ser259-MYRF can serve as potentially therapeutic targets for mitigating demyelination in HD, and perhaps in other diseases with demyelination as well.

# Materials and Methods

### Animals

The generation of transgenic PLP-23Q and PLP-150Q mice that express N-terminal human HTT (amino acids 1–212) containing 23Q or 150Q was described in our previous studies [23]. Genomic DNA was isolated from mouse tails, and PCR genotyping with primers flanking the polyCAG region (Table 1) was performed to identify transgenic mice.

PLP-GFP mice were obtained from Dr. Wendy Macklin (University of Colorado Denver) and bred with PLP-150Q mice to generate double-transgenic mice. Full-length mHtt CAG140Q (HD-KI) mice were described in our previous study [23]. PLP-150Q mice were crossed with Cas9 transgenic mice, which were obtained from the Jackson Laboratory and described previously [38]. The crossed mice expressing mutant HTT and Cas9 were verified by amplifying PCR fragment using the primers of Cas9-WT and Cas9-Mutant in Table 1.

All mice were bred and maintained in the animal facility at Emory University under specific pathogen-free conditions in accordance with the institutional guidelines of the Animal Care and Use Committee at Emory University.

### Reagents

Commercial antibodies used in this study include the following: 1C2 (Millipore, MAB1574), GAPDH (Millipore, MAB374), MBP (Millipore, MAB386), MOBP (Santa Cruz, sc-14520), MOG (Millipore, MAB5680), MYRF (Sigma, HAP018310), PRKG2 (Proteintech, 55138-1-AP), phospho-PRKG2 (Invitrogen, PA5-64606), phosphoserine (Chemicon, AB1603), AhR (Enzo, BML-SA210-0100), phospho-VASP (Cell Signaling, 3114S), and Olig2 (Millipore, AB9610). Rabbit polyclonal antibody (EM48) and mouse monoclonal antibody (mEM48) against the N-terminal region (amino acids 1-256) were generated as described in our previous study [57]. Rabbit polyclonal antibody of phospho-S259-MYRF (J6738) was generated with a synthetic peptide containing amino acids 256-269 (RKH pSESPPNTLNA) of mouse MYRF as antigen, which were generated by New England Peptide Inc. Company (Gardner, MA, USA). Lambda protein phosphatase was from New England Biolabs (P0753S); 8-Br-cGMP (15992), KT5823 (10010965), and RKRARKE (15995) were from Cayman Chemical (Ann Arbor, MI, USA).

PRK plasmids expressing HA-tagged N-terminal huntingtin (amino acids 1–212) (PRK-HTT-23Q or -150Q) and DNA construct for expressing N-terminal MYRF (nMYRF) were generated as described previously [23]. The nMYRF-Myc cDNA containing a Myc tag was generated using the primer pairs of nMYRF-Myc in Table 1. GST-tagged nMYRF was generated using the PCR primers of nMYRF-GST in Table 1. The PCR product was inserted into the PGEX4T-1 vector at EcoR1 and Sal1 restriction sites. To generate mutant MYRF with S259A and S261A, the Q5-Site-Directed Mutagenesis Kit (New England Biolabs) with the primer pairs of nMYRF-S259A and nMYRF-S261A in Table 1 were used to create mutations. For the luciferase reporter assay, the DNA fragment containing MBP promoter was amplified from mouse genomic DNA using the PCR primers of promoter-MBP. The promoter DNA was then inserted into the luciferase reporter vector pGL4.14 (Promega) at KpnI and HindIII restriction sites. Mouse PRKG2 promoter DNA containing different lengths were generated using the following PCR primers of PRKG2-1209, PRKG2-866, PRKG2-472, and PRKG2-161 in Table 1, inserted into the luciferase reporter vector pGL4.14 at Xhol and HindIII restriction sites. To generate an overexpression mouse PRKG2 vector, the following

**Table 1. Primer sequences for plasmid constructs, qPCR, and ChIP assay.**

| Name | Forward primer | Reversed primer |
| --- | --- | --- |
| PLP-polyCAG | 5′-ATGAAGGCCTTCGAGTCCCTCAAGTCCTTC-3′ | 5′-AAACTCACGGTCGGTGCAGCGGCTCCTCAG-3′ |
| Cas9-WT | 5′-AAGGGAGCTGCAGTGGAGTA-3′ | 5′-TCGAAAATCTGTGGGAAGTC-3′ |
| Cas9-Mutant | 5′-AAGGGAGCTGCAGTGGAGTA-3′ | 5′-TGGGCCATTTACCGTAAGTTAT-3′ |
| nMYRF-Myc | 5′-ACATCGATACCATGGAACAAAAGCTCATTAGCGAGGA AGACCTTATGGAGGTGGTGGACGAGAC-3′ | 5′-TCTTGTCGACCTAGGTGGTGTCCACCTCCTGCACGTGCTC-3′ |
| nMYRF-GST | 5′-AGAATTCGCCATGGAGGTGGTGGACGAGACCGAAGCGCTGCAG-3′ | 5′-TCTTGTCGACCTAGGTGGTGTCCACCTCCTGCACGTGCTC-3′. |
| nMYRF-S259A | 5′-AAGAGGAAGCACGCAGAATCACCCCCCCAAC-3′ | 5′-CTTAGAGGGGTGTGGAGGGAGTTCAGCTC-3′ |
| nMYRF-S261A | 5′-AAGCACTCTGAAGCACCCCCCAACACCCTC-3′ | 5′-CCTCTTCTTAGAGGGGTGTGGAGGGAGTT-3′. |
| Promoter-MBP | 5′-ATGGTACCCTGTGTGAGCATGTGACA-3′ | 5′-TTTAAGCTTGAACAGTCCCCGTGAGG-3′. |
| PRKG2-1209 | 5′-CCGCTCGAGTTTTACCTCAATGGACAGAGAGTAC-3′ | 5′-CCCAAGCTTAGCTCAGCGGAGAGGAGGGACGGAGG-3′ |
| PRKG2-866 | 5′-CCGCTCGAGTCATTCTATGTGGGGTTATGTATGT-3′ | 5′-CCCAAGCTTAGCTCAGCGGAGAGGAGGGACGGAGG-3′ |
| PRKG2-472 | 5′-CCGCTCGAGGAATCAGGAGGTATTAATTCCTGTC-3′ | 5′-CCCAAGCTTAGCTCAGCGGAGAGGAGGGACGGAGG-3′ |
| PRKG2-161 | 5′-CCGCTCGAGATGATGAATACCCCCCCTCAACTTTC-3′ | 5′-CCCAAGCTTAGCTCAGCGGAGAGGAGGGACGGAGG-3′ |
| PRKG2-cDNA | 5′-CCGGAATTCATGGGAAATGGTTCAGTGAAGCCCAAGCATG-3′ | 5′-CCCAAGCTTTCAGAAGTCTTTATCCCAGCCTGACATTTC-3′ |
| MBP-qPCR | 5′-AGTACCTGGCCACAGCAAGT-3′ | 5′-AGGATGCCCGTGTCTCTGT-3′ |
| MOG-qPCR | 5′-ACGGGCATGGAGGTGGGGTT-3′ | 5′-CAGGTGCTTGCTCTGCATCTTG-3′ |
| GAPDH-qPCR | 5′-AACTTTGTCAAGCTCATTTCCTGGT-3′ | 5′-GGTTTCTTACTCCTTGGAGGCCATG-3′ |
| Cyp1a1-qPCR | 5′-CGTGAGCAAGGAGGCTAACTAT-3′ | 5′-TGGCTACTGACACGACCAAATAC-3′ |
| Ugt1a6a-qPCR | 5′-GTCCCTACCTTCCTTACACCAG-3′ | 5′-TCAGCTTTCCCTTCTTCTTACAG-3′ |
| PRKG1-qPCR | 5′-ACCCTTGGAGTTGGAGGTTTCG-3′ | 5′-TGGCGTTTCTTGAGGATCTTCATT-3′ |
| PRKG2-qPCR | 5′-TACGGGAGAAACTATCAACAGG-3′ | 5′-GGAACACCTCAAGTCTACCCTC-3′ |
| AhR-ChIP-1 | 5′-AGTCCTAGAAGAGCACGCGGGCCA-3′ | 5′-GTGGGCTCTTGCTCTCCCTTGCT-3′ |
| AhR-ChIP-2 | 5′-CCTCTCCCTCGGCCCACACACTTG-3′ | 5′-GGCGACGGCGACAGTCTGCACTTTA-3′ |
| AhR-ChIP-34 | 5′-CCCCACCTTCAAGGAAGCGGGCT-3′ | 5′-AGCTCAGCGGAGAGGAGGGACGGA-3′ |
| Cas9-PRKG2 | 5′-ACTGCCGTGTGTCTGTTTATGAAGC-3′ | 5′-CAGCATCCAGTGGCAATTCCAGATC-3′ |

PCR primers were used to amplify the PRKG2 cDNA according to NCBI published sequence; the full-length PRKG2 open-reading frame was inserted into the AAV-MCS vector at EcoR1 and HindIII restriction sites, using the primers of PRKG2-cDNA in Table 1. To generate the AAV-U6-gRNA-CMV-RFP vector for mouse PRKG2 in CRISPR/Cas9 knocking down experiment, the following gRNAs were expressed in the PX552-RFP vector (Addgene), T1: 5′-GAT GTGATCCACGTGCAGGGAGG-3′; and T2: 5′-CTGGATGTTCACCG CAAGAC-3′.

**Administration of LAQ**

LAQ was provided by Teva Pharmaceutical Industries and was dissolved in sterile water. LAQ and vehicle were administered into mice by oral gavage daily starting at 3 months of age, for a period of 2 months. Mice received vehicle (sterile water), 5 or 25 mg/kg of LAQ at a volume of 4 ml/kg.

**Mouse behavior analysis**

Because aging can influence HD pathology, we used age-matched control animals. Age-matched control animals and LAQ-treated animals were randomly grouped. However, the identifies of the grouped animals were blinded to investigators when analyzing their behaviors. Mouse body weight was measured once every 3 days, and survival was monitored regularly. The motor function of the mice was assessed with the rotarod test (Rotamex; Colum bus Instruments). Mice were trained on the rotarod at 5 rpm for 5 min for 3 consecutive days. After training, the mice were tested for 3 consecutive days, 3 trials per day. The rotarod was gradually accelerated to 40 rpm over a 5-min period. Latency to fall of mice was recorded for each trial. The balance beam apparatus consists of 1-meter beams with a flat surface of 12 mm or 6 mm width resting 50 cm above the tabletop on two poles. A black box at the end of the beam is the finish point. Nesting material from home cages in the black box served to attract the mouse to the finish point. A lamp served as an aversive stimulus, shining light above the start point. The time required for a mouse to cross to the center (80 cm) was measured by two motion detectors: one at 0 cm that starts a timer and one at 80 cm that stops the timer. The video camera recorded the mouse performance. At least 12 mice were analyzed per group when comparing vehicle and LAQ treatment on PLP-150Q or PLP-23Q mouse.

**Analysis of oligodendrocyte morphology**

To examine oligodendrocyte morphology, we used PLP-150Q/GFP mice that also express GFP in oligodendrocytes. After perfusion with 4% paraformaldehyde (PFA) in 0.1 M phosphate buffer (PB),

mouse brains were cut into 18-μm sections with a cryostat. Images were acquired with a Zeiss microscope (Carl Zeiss Imaging, Imager A.2), digital camera (Carl Zeiss Imaging, AxioCam HRc), and 40× lens (LD-Achroplan 40×/0.6). Using the Simple Neurite Tracer plugin for ImageJ, lengths of GFP-positive processes in the striatum and cortex were measured from the cell body to the tip of the process. Lengths were reported in arbitrary units (AU), which are defined as the pixel values from 8-bit luminance images used for quantification. Three 3-month-old mice were examined for each genotype, and at least 148 cells per genotype were analyzed. Images were acquired with a Zeiss microscope (Carl Zeiss Imaging, Imager A.2), digital camera (Carl Zeiss Imaging, AxioCam HRc), and 10× lens (EC Plan-Neufluar 10×/0.3). The Cell Counter plugin for ImageJ was used to count the number of GFP-positive oligodendrocytes per 10× field.

### In vitro phosphorylation assay

Purified GST-nMYRF linked on sepharose beads were diluted in ice-cold assay buffer (25 mM Tris–HCl, 10 mM $MgCl_2$, 100 μg/ml purified rabbit creatine kinase, 50 mM phosphocreatine, 1 mM ATP, pH 7.6). Mouse tissues from the corpus callosum were homogenized at 1 g/ml in ice-cold assay buffer using 20 strokes of a glass Dounce homogenizer, and the homogenates were centrifuged at 500× *g* for 5 min at 4°C to pellet unbroken tissues and membranes. The supernatant was collected and stored on ice while protein concentrations were determined using a BCA Protein Assay Kit (Thermo Scientific/Pierce). The lysates (200 μl) at 0.5 mg protein/ml were incubated with GST-nMYRF beads (200 μl) at 37°C for 1 h with shaking at 300 rpm. The beads were centrifuged and combined with the protein loading dye (0.2% SDS) for SDS–PAGE and Western blot analysis to detect phosphorylated proteins using anti-phosphorserine or anti-GST antibody.

### Transfection of cultured cells

Human embryonic kidney (HEK) 293 cells or mouse neuroblastoma Neuro-2a cells (N2a) were purchased from ATCC and cultured in DMEM/F12 containing 10% fetal bovine serum, 100 U/ml penicillin, and 100 μg/ml streptomycin (Invitrogen). Medium was changed every 2 days. For immunofluorescent staining of transfected HEK293 cells, cells were transfected with N-terminal MYRF alone or with N-terminal (1–212 amino acids) HTT-150Q using Lipofectamine-2000 transfection reagent (Invitrogen) according to the manufacturer's protocol. For siRNA knockdown experiment, the cultured mouse N2a cells were transiently transfected with AhR siRNA (Gene Pham Co.) of the sequence GAGGGAUUAACUUCUA GAUtt and AUCUAGAAGUUAAUCCCUCUtt [35] or control siRNA (scrambled sequence) using RNAi Max transfection reagent (Invitrogen) according to the manufacturer's protocol. At 48 h following transfection, cells were harvested for immunofluorescent staining and Western blotting.

### Immunofluorescence, immunohistochemistry, and electron microscopy

Mice were anesthetized with 5% chloral hydrate and perfused with 0.9% NaCl, followed by 4% paraformaldehyde (PFA). The brains were removed and post-fixed in 4% PFA overnight at 4°C. The brains were transferred to 30% sucrose for 48 h and then cut to 20- or 40-μm sections with the cryostat (Leica CM1850) at −20°C. Sections were blocked in 4% donkey serum with 0.2% Triton X-100 and 3% BSA in PBS for 1 h. For immunofluorescent staining, 20-μm sections were incubated with primary antibodies in the same buffer at 4°C overnight. After washing with 1× PBS, the sections were incubated in fluorescent secondary antibodies. Fluorescent images were acquired with a Zeiss microscope (Carl Zeiss Imaging, Axiovert 200 MOT) and either a 40 or 63× lens (LD-Achroplan 40×/0.6 or 63×/0.75) with a digital camera (Hamamatsu, Orca-100) and Openlab software (Improvision). For immunohistochemistry with DAB staining, after blocking, 40-μm sections were incubated with antibody for at least 48 h at 4°C. A biotin/avidin immuno-assay (Vector Laboratories) and DAB kit (Invitrogen, 00-2020) were used. Images were acquired with a Zeiss microscope (Carl Zeiss Imaging, Imager A.2) and either a 40 or 63× lens (Plan-Apochromat 40×/0.95 or 63×/1.4) with a digital camera (Carl Zeiss Imaging, AxioCam HRc) and AxioVision software. For electromicroscopic study, mice were perfused with 0.9% NaCl followed by 4% PFA containing 2.5% glutaraldehyde. After post-fixation, the brain was cut to 50-μm sections using the vibratome (Leica, VT1000). Axon and myelin fiber diameters were measured using ImageJ (NIH). More than 300 axons were examined for each genotype.

### Western blotting and co-immunoprecipitation

To examine myelin protein levels, brain tissues were dissected out and sonicated three times for 10 s in 1× SDS sample buffer and 0.5% Triton X-100 in 1× PBS containing protease inhibitor mixture (Roche) and 100 μM PMSF. The samples were incubated at 100°C for 10 min and centrifuged at 4°C for 10 min at 16,000× *g*. Samples were resolved in a 4–20% Tris-glycine SDS–polyacrylamide gel (Invitrogen) and transferred onto a nitrocellulose membrane. Blots were developed with ECL Prime (GE Healthcare, RPN2232). ImageJ was used for Western blot quantification. For co-immunoprecipitation using mouse brain tissue, total 500 μg tissues of corpus callosum in cold 1% NP-40 buffer (50 mM Tris, pH 7.4, 50 mM NaCl, 0.1% Triton X-100, and 1% NP-40 with 1× protease inhibitors and 100 μM PMSF) were precleared with protein A agarose beads (Sigma, P1406); then, the samples were immunoprecipitated by anti-Htt (mEM48) at 4°C overnight. Protein A agarose beads were added to capture the immunoprecipitates for 2 h at 4°C. Ice-cold lysis buffer was used to wash beads three times. Proteins from the immunoprecipitates and inputs were subjected to Western blotting. For immunoprecipitation, the HEK293 cells were co-transfected with MYRF and HTT containing 23Q or 150Q. Forty-eight hours following transfection, the cells were harvested in cold 1% NP-40 buffer. Samples were precleared and incubated overnight with either MYRF or EM48 antibody at 4°C. The mass spectrometry for post-translational modifications (PTMs) analysis for the phosphorylation signals on MYRF was performed by Emory Integrated Proteomics Core (EIPC). The immunoprecipitated N-terminal MYRF band was confirmed by Western blotting, and the corresponding band in the Coomassie blue R250-stained gel was isolated for the mass spectrometry analysis.

## Microarray and qRT–PCR

For the microarray assay, total RNA was isolated from the mouse corpus callosum with the RNeasy Lipid Tissue Mini Kit (Qiagen) from 4 Vehicle and 4 LAQ-treated PLP-150Q mice. The Mouse Gene 1.0 ST Array (Affymetrix) was used. Probe sets with at least 1.5-fold change and ANOVA $P < 0.05$ were considered significant. Data were subsequently subjected to ingenuity pathway analysis (Ingenuity Systems). For qRT–PCR, total corpus callosum RNA was isolated from 4 5-month-old mice of each genotype after LAQ treatment. Reverse transcription reactions were performed with 1.5 μg of total RNA using the Superscript III First-Strand Synthesis System (Invitrogen, 18080-051). One microliter of cDNA was combined with 10 μl SYBR Select Master Mix (Applied Biosystems, 4472908) and 1 μl of each primer in a 20 μl reaction. The reaction was performed in a real-time thermal cycler (Eppendorf, Realplex Mastercycler). The PCR products were analyzed on a 2% agarose gel to ensure that the PCR amplification only produced a single specific band. The sequences of the primers of MBP-qPCR, MOG-qPCR, GAPDH-qPCR, Cyp1a1-qPCR, Ugt1a6a-qPCR, PRKG1-qPCR, and PRKG2-qPCR are in Table 1. The GAPDH served as an internal control. Relative expression levels were calculated using $2^{-\triangle\triangle CT}$, with WT set at 1.

## Luciferase reporter assay

HEK293 cells were transfected with PRK-nMYRF, N-terminal HTT (1–212 amino acids) PRK-HTT-150Q or PRK-HTT-23Q and pGL4.1-pMBP (MBP promoter luciferase). After 48 h, cells were harvested and lysed with lyses buffer. Each sample was divided into duplicates, one half was lysed in 1% NP-40 buffer, and protein levels were determined via Western blotting. The remaining half was used for the reporter assay using Luciferase Assay System (Promega, E1483), and luciferase activity was detected with a luciferase reader (BioTek, Synergy H4). For the rescue experiment, HEK293 cells were transfected with different combinations of the following plasmids: MBP promoter luciferase, PRK-HTT-150Q or PRK-HTT-23Q and nMYRF, with the treatment of LAQ added.

## Stereotaxic injection of AAV viruses

AAV viruses (type-9) were packaged and amplified by the Viral Vector Core at Emory University. The titers of AAV vector genome were $6.9 \times 10^{12}$ vg/ml for AAV-mouse-PRKG2, $2.3 \times 10^{13}$ vg/ml for AAV-GFP, $1.8 \times 10^{14}$ vg/ml for AAV-gRNA-control, and $3.6 \times 10^{13}$ vg/ml for AAV-gRNA-PRKG2. Adult wild-type C57BL/6 mice at 2 or 3 months of age ($n = 6$ each group, three males and three females per group) were anesthetized by i.p. injection of 2.5% avertin, and their heads were placed in a Kopf stereotaxic frame (Model 1900) equipped with a digital manipulator, a UMP3-1 Ultra pump, a 10-μl Hamilton microsyringe. A 33-G needle was inserted through a 1-mm drill hole on the scalp. Injections occurred at the following stereotaxic coordinates: 0.6 mm prior to bregma and 2.5 mm ventral to the dura, with bregma set at zero, abiding by the midline for the middle corpus callosum, while 1.5 mm lateral to the midline for the lateral corpus callosum. The microinjections were carried out at a rate of 0.2 μl/min. The microsyringe was left in place for an additional 10 min before and after each injection, and 0.5 μl AAV virus was stereotaxically injected into both the middle and lateral corpus callosum of mice.

## T7 endonuclease 1 assay and TA cloning

N2a cells in a six-well plate were co-transfected with PRKG2 gRNA and Cas9 plasmids or transfected only with PRKG2 gRNA as control using Lipofectamine 2000 (Invitrogen). Seventy-two hours after transfection, the genomic DNA was extracted from transfected N2a cells. The target genomic region was amplified with PCR using the primers of Cas9-PRKG2 in Table 1. The PCR products were denatured and reannealed and then incubated with T7 Endonuclease I (New England BioLabs) for 20 min at 37°C. The reaction products were subjected to 2% agarose gel electrophoresis. The PCR product was inserted into pCR2.1-TOPO vector using vector TOPO TA Cloning Kit for sequencing.

## Human tissue acquisition

Human cortex tissues were obtained and archived via an institutional review board and Health Insurance Portability and Accountability Act-compliant process at Harvard Brain Tissue Resource Center (HBTRC, Belmont, MA, USA). Autopsies occurred following death with post-mortem interval of 6–13 h. The cortex tissues were obtained from the post-mortem brains of HD patients (B4094, B5259, B5297, and B5299) and were confirmed via post-mortem histologic analysis. Non-HD control tissues were obtained from neurologically unaffected patients (B4106, B4131, B4174, and B4193).

## Statistical analysis

Results are expressed as mean ± SEM. Prism6 (GraphPad Software) was used for statistical analysis. When only two experimental groups were compared, Student's *t*-test was used to calculate statistical significance. For all other experiments, statistical significance was calculated using one-way ANOVA or two-way ANOVA, followed by Tukey's multiple-comparisons test. A $P < 0.05$ was considered significant.

# Data availability

The raw data from this publication have been deposited. The RNA sequencing data that support the findings of this study are available in NCBI's Gene Expression Omnibus and are accessible through GEO Senes reference GSE145044 (https://www.ncbi.nlm.nih.gov/geo/query/acc.cgi?acc=GSE145044).

**Expanded View** for this article is available online.

# Acknowledgements

This work was supported by a Research Contract from Teva Pharmaceuticals and grants from the NIH (NS095181) and National Natural Science Foundation of China (81600990, 81830032, 31872779). We also thank Hong Yi at Emory Electron Microscopy Core (supported by the Georgia Clinical and Translational Science Alliance under award number UL1TR002378) for electron microscopic

analysis, Tharp Gregory at Emory Yerkes National Primate Research Center for processing RNAseq raw data, Duong Duc at Emory Integrated Proteomics Core (supported by NIH/NCI under award number P30CA138292) for the mass spectrometry analysis and Xinping Huang at Emory Viral Vector Core (supported by Neuroscience NINDS Core Facilities grant P30NS055077) for virus packaging and purification. Human tissue samples were provided by the Harvard Brain Tissue Resource Center, which is supported in part by NIH NeuroBioBank and by Marla Gearing and Jonathan D. Glass from Emory Alzheimer's Disease Center Brain Bank (ADRC grant of NIA P50 AG025688).

## Author contributions

X-JL, PY, and SL designed experiments. PY, QL, YP, YH, and DB performed experiments. PY, WY, SY, WW, and XC analyzed data. X-JL, PY, and SL wrote the paper.

## Conflict of interest

The authors declare that they have no conflict of interest.

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
