## [Review Process File · EMBO Reports]

Phosphorylation of myelin regulatory factor by PRKG2 mediates demyelination in Huntington's disease

Peng Yin, Qiong Liu, Yongcheng Pan, Weili Yang, Su Yang, Wenjie Wei, Xingxing Chen, Yan Hong, Dazhang Bai, Xiao-Jiang Li and Shihua Li

Review timeline:	Submission date:	29 November 2019
	Editorial Decision:	20 January 2020
	Revision received:	26 January 2020
	Editorial Decision:	10 March 2020
	Revision received:	10 March 2020
	Accepted:	12 March 2020

Editor: Esther Schnapp

Transaction Report:

1st Editorial Decision

20 January 2020

Thank you for your patience while your manuscript was peer-reviewed at EMBO reports. We have finally received the full set of referee reports that is pasted below.

As you will see, while referee 1 is more critical, both referees 2 and 3 agree that the findings are interesting and should be published, and that only minor revisions are required. However, I would like to suggest that you also address referee 1's concerns to the best of your abilities. If you have any questions regarding the revisions, please contact me.

I would thus like to invite you to revise your manuscript with the understanding that the referee concerns must be fully addressed and their suggestions taken on board. Please address all referee concerns in a complete point-by-point response. Acceptance of the manuscript will depend on a positive outcome of a second round of review. It is EMBO reports policy to allow a single round of major revision only and acceptance or rejection of the manuscript will therefore depend on the completeness of your responses included in the next, final version of the manuscript.

Revised manuscripts should be submitted within three months of a request for revision; they will otherwise be treated as new submissions. Please contact us if a 3-months time frame is not sufficient for the revisions so that we can discuss this further.

Regarding data quantification, please specify the number "n" for how many independent experiments were performed, the bars and error bars (e.g. SEM, SD) and the test used to calculate p-values in the respective figure legends. This information must be provided in the figure legends. Please also include scale bars in all microscopy images.

1) Your manuscript contains statistics and error bars based on n=2 or on technical replicates. Please

use scatter blots in these cases. No statistics can be calculated if $n=2$.

3) We replaced Supplementary Information with Expanded View (EV) Figures and Tables that are collapsible/expandable online. A maximum of 5 EV Figures can be typeset. EV Figures should be cited as 'Figure EV1, Figure EV2' etc... in the text and their respective legends should be included in the main text after the legends of regular figures.

5) a complete author checklist, which you can download from our author guidelines <https://www.embopress.org/page/journal/14693178/authorguide>. Please insert information in the checklist that is also reflected in the manuscript. The completed author checklist will also be part of the RPF.

6) Please note that all corresponding authors are required to supply an ORCID ID for their name upon submission of a revised manuscript (<https://orcid.org/>). Please find instructions on how to link your ORCID ID to your account in our manuscript tracking system in our Author guidelines <https://www.embopress.org/page/journal/14693178/authorguide#authorshipguidelines>

7) We would also encourage you to include the source data for figure panels that show essential data. Numerical data should be provided as individual .xls or .csv files (including a tab describing the data). For blots or microscopy, uncropped images should be submitted (using a zip archive if multiple images need to be supplied for one panel). Additional information on source data and instruction on how to label the files are available at <https://www.embopress.org/page/journal/14693178/authorguide#sourcedata>.

8) Our journal also encourages inclusion of *data citations in the reference list* to directly cite datasets that were re-used and obtained from public databases. Data citations in the article text are distinct from normal bibliographical citations and should directly link to the database records from which the data can be accessed. In the main text, data citations are formatted as follows: "Data ref: Smith et al, 2001" or "Data ref: NCBI Sequence Read Archive PRJNA342805, 2017". In the Reference list, data citations must be labeled with "[DATASET]". A data reference must provide the database name, accession number/identifiers and a resolvable link to the landing page from which the data can be accessed at the end of the reference. Further instructions are available at <https://www.embopress.org/page/journal/14693178/authorguide#referencesformat>

I look forward to seeing a revised version of your manuscript when it is ready. Please let me know if you have questions or comments regarding the revision.

REFeree REPORTS

Referee #1:

The authors show that LAQ treatment reduces the Ser259 phosphorylation on MYRF and increases the activity of PRKG2 in the CNS of PLP-150Q mice. They further show that PRKG2 promotes the Ser259-MYRF phosphorylation and that knocking-down PRKG2 increases the activity of MYRF in the CNS of PLP-150Q mice. Based on these findings, the authors conclude that phosphorylation of MYRF by PRKG2 mediates demyelination in Huntington's disease. Although the topic is potentially interesting, the manuscript is written very poorly and the data is very rough and unconvincing. There are a number of major problems.

1. All the immunostaining data are extremely rough. It is unclear what the authors want to show.
2. There is no evidence Ser259 phosphorylation on MYRF, or alteration of PRKG2 activity is occurred in oligodendrocytes in the mouse model or human samples. LAQ has a very broad effect on the CNS. The authors should demonstrate that these effects occur selectively in oligodendrocytes.
3. Only oligodendrocytes produce myelin in the CNS. To determine the role of MYRF or PRKG2 in the myelinating function of oligodendrocytes, the authors should use oligodendrocytes, rather than HEK293 cells or N2a cells. The data from HEK293 cells or N2a cells have little or no relevance to oligodendrocytes.
4. There is no evidence showing that knocking-down PRKG2 attenuates demyelination in the CNS of PLP-150Q mice. The authors should perform EM analysis to demonstrate this major finding.

Referee #2:

This is an important follow up of a 2016 paper in *Neuron*, which reports the association of Huntingtin (HTT) with the promyelinating transcription factor MRF in oligodendrocytes. A transgenic PLP-HTT(150Q) mouse model was generated at the time that showed the occurrence of a primary "demyelination" in Huntington disease. Now the authors provide compelling evidence that a specific phosphorylated form of MYRF (at Ser259) binds to mutant HTT. Laquinimod is a polyaromatic compound that had shown significant success in the treatment of demyelination in both human multiple sclerosis and Huntington disease, but its mechanism of action is poorly understood. Here, the authors discover that Laquinimod interferes with the phosphorylation of HTT. In consequence, Laquinimod treatment renders MYRF less likely to be bound by mutant HTT, which in turn improves myelin gene expression and the motor phenotype of PLP-HTT(150Q) mice, but not the early lethal outcome. Expectedly, also knocking-down the identified responsible kinase (PRKG2) increased myelin gene expression in PLP-HTT mice. The authors can confirm their model by showing direct protein interactions.

This is a very interesting set of observation, a comprehensive analysis and first mechanistic insight into HTT-MRF interactions. As the authors discuss, there are likely additional cytotoxic effects of mutant HTT, as the rescue of oligodendroglial dysfunction in PLP-HTT mice is incomplete. In turn, there must be HTT-independent effects of this drug, because the successful treatment of

demyelination in MS patients is in the absence mutant HTT expression. Nevertheless, this paper marks the first molecular insight into the obscure role of Laquinomod as a drug that works in human patients. There is certainly room for improvement of some of the figures, including the summary diagram, but I have no major problems with this paper as is. My only concern is the diagnosis of the mutant PLP-HTT mutant phenotype as "demyelination", without showing the activity of macrophages and low grade inflammation. More likely the thinly myelinated axons reflect a developmental problem (dysmyelination), which is nevertheless interesting.

Referee #3:

The Ms by Yin and colleagues investigate the mechanisms by which Laquinimod (LAQ), a molecule therapeutic interest in MS and HD restores myelination. Indeed, although several studies have described the protective effect of this drug, the molecular mechanism remain elusive. Here, Yin and colleagues, using a mouse model of HD that selectively express mutant huntingtin (mHTT) in oligodendrocytes and that show demyelination, show that LAQ reduce the level of phosphorylation of the myelin regulatory factor. Authors identified the site of phosphorylation as being S259 by performing MS on immunopurified MYRF. In HD mice, MYRF phosphorylation is increased thus leading to the interaction between MYRF and HTT and subsequent reduction of MBP transcription. Reducing S259 phosphorylation reduces MYRF-HTT interaction and restores transcriptional activity at the MBP promoter. Authors then raised a S259 P specific antibody and found that MYRF phosphorylation is increased in vivo and in HD patients and that phosphorylation is reverted in vivo after LAQ treatment in mice.

How LAQ leads to MYRF S250 phosphorylation? By performing RNA seq on mice treated by LAQ, authors identify PRKG2, a cGMP-activated protein kinase subunit II as being downregulated. By searching for the potential transcription factors that can bind to the PRKG2, authors identify binding sites of Aryl hydrocarbon receptor on PRGK2 promoter and using ChIP assay, found that LAQ acts on Aryl hydrocarbon receptor by increasing its association with the PRGK2 promoter. Next they show that PRGK2 effectively phosphorylate S259 of MYRF both in vitro in cells and in vivo in mouse brain. Finally, authors used CRISPR/Cas9 approach to knock-out PRGK2 in the corpus callosum of PLP-150Q HD mice and observed a reduction of MYRF phosphorylation and increase in MBP expression.

Together, the authors describe a molecular mechanism by which LAQ acts to increase myelination. These findings with the identification of PRGK2 as a mediator of LAQ therapeutic effect is of interest not only for HD but for other disorders with demyelination.

This is a very complete study that should be published as it is given the quality of the experiments and the importance of the findings.

1st Revision - authors' response

26 January 2020

Referee #1:

The authors show that LAQ treatment reduces the Ser259 phosphorylation on MYRF and increases the activity of PRKG2 in the CNS of PLP-150Q mice. They further show that PRKG2 promotes the Ser259-MYRF phosphorylation and that knocking-down PRKG2 increases the activity of MYRF in the CNS of PLP-150Q mice. Based on these findings, the authors conclude that phosphorylation of MYRF by PRKG2 mediates demyelination in Huntington's disease. Although the topic is potentially interesting, the manuscript is written very poorly and the data is very rough and unconvincing. There are a number of major problems.

1. All the immunostaining data are extremely rough. It is unclear what the authors want to show.

We have modified title of figures and also improved the clarity of the statements in the text for the immunostaining data. We also used better quality images in the revised figures.

2. There is no evidence Ser259 phosphorylation on MYRF, or alteration of PRKG2 activity is

occurred in oligodendrocytes in the mouse model or human samples. LAQ has a very broad effect on the CNS. The authors should demonstrate that these effects occur selectively in oligodendrocytes.

The reviewer raised an important issue about the selective effect in oligodendrocytes observed in our study. We believed that the transgenic mouse model we studied and the selective expression of MYRF in oligodendrocytes allowed us to investigate the selective effect of mutant Htt and MYRF in oligodendrocyte. First, PLP-150Q mice selectively express mutant Htt in oligodendrocytes, which has been shown in our previous study (Huang et al., Neuron 2015), as this mouse model expresses mutant Htt under the control of the oligodendrocyte specific promoter (PLP). Second, MYRF is a transcription factor that is selectively expressed in oligodendrocytes and activate the expression of oligodendrocyte specific genes such as MBP (Emery et al., Cell 2009; Bujalka et al., PLoS Biol. 2013). Thus, although LAQ has broad effects, the altered Ser259 phosphorylation of MYRF and MBP should be oligodendrocyte specific. Similarly, when we examined MBP expression changes after altering PRKG2 activity, the altered expression of MBP also reflects the effect of PRKG2 in oligodendrocytes because MBP is only expressed in oligodendrocytes.

In addition, we compared PLP-23Q and PLP-150Q mice, which all express Htt with different repeats in oligodendrocytes. The difference between PLP-23Q and PLP-150Q also reflects the selective effect of mutant Htt in oligodendrocytes. Thus, using HD mice that selectively express Htt in oligodendrocytes to investigate the oligodendrocyte-specific gene expression and function would enable us to identify the selective effect of LAQ in oligodendrocytes.

In the revised summary diagram in Figure 7, we also indicate the specific effect of LAQ in oligodendrocytes in HD. In addition, we included the following discussion in the revised text on page 12:

“In the current studies, we examined the effect of LAQ in PLP-150Q mice that selectively express mutant Htt in oligodendrocytes. We also compared PLP-150Q mice with PLP-23Q mice that express normal Htt in oligodendrocytes. Thus, the alleviation of the phenotypes of PLP-150Q mice by LAQ should be specifically related to the effect of LAQ on mutant Htt in oligodendrocytes. In addition, we focused on MYRF and MYRF-mediated expression of myelin genes such as MBP. Since MYRF and MBP are only expressed in oligodendrocytes and since the expression of mutant Htt in PLP-150Q mice is restricted to oligodendrocytes, we could identify the selective effects of mutant Htt and LAQ treatment in oligodendrocytes by comparing with PLP-23Q mice that express normal N-terminal Htt in oligodendrocytes.”

3. Only oligodendrocytes produce myelin in the CNS. To determine the role of MYRF or PRKG2 in the myelinating function of oligodendrocytes, the authors should use oligodendrocytes, rather than HEK293 cells or N2a cells. The data from HEK293 cells or N2a cells have little or no relevance to oligodendrocytes.

In Figure 6 and 7, the majority of data describe the effect of PRKG2 on myelin protein MBP in the mouse brain. We also developed a unique antibody for the phosphorylated MYRF and provided a considerable amount of in vivo data to show the phosphorylation of MYRF in oligodendrocytes in the brains of HD mice and patients (see Figure 3). Because MYRF and myelin are only expressed in oligodendrocytes, the data represent the functional changes that selectively occur in oligodendrocytes.

Only for the molecular characterization of the interaction of MYRF and mutant Htt, we used transfected cells, because we need to use different forms of MYRF (full-length, truncated, phosphorylated and non-phosphorylated) to examine their interactions with normal and mutant Htt. Such in vitro studies have been widely used for investigating protein-protein interaction and phosphorylation. For example, many high quality journals published in vitro studies of MYRF and its interaction protein in HEK293 or N2a cell line (Meng et al., Dev Cell; Bujalka et al., PLoS Biol. 2013; Hornig et al., PLoS Genet. 2013; Huang et al Neuron. 2015).

In addition, oligodendrocyte culture is very difficult, because these cells are difficult to be purified and survive in vitro. Our in vitro studies were to analyze the interaction and

phosphorylation of MYRF, which were then validated by the substantial amount of *in vivo* evidence (Fig 5D-F and Fig. 6).

4. There is no evidence showing that knocking-down PRKG2 attenuates demyelination in the CNS of PLP-150Q mice. The authors should perform EM analysis to demonstrate this major finding.

We appreciate this good suggestion. We had injected a limited number of PLP-150Q mice with AAV-PRKG2 gRNA and wanted to obtain evidence for the effect of altered phosphorylation of MYRF on MBP. Thus, we used the injected mouse brains for western blotting and immunostaining of MBP. This is because MYRF regulates the expression of MBP such that changes in MBP level reflect MYRF function that can be modulated by knocking down PRKG2.

Although the suggestion of performing EM to examine demyelination is good, alteration of myelin structure and morphology can be caused by multiple factors, not necessarily indicating the change in MYRF function. In addition, we had difficulty now obtaining more PLP-150Q mice to perform this experiment, as this experiment needs to use new PLP-150Q mice and core facility, which are not available to us anymore.

Referee #2:

This is an important follow up of a 2016 paper in *Neuron*, which reports the association of Huntingtin (HTT) with the promyelinating transcription factor MRF in oligodendrocytes. A transgenic PLP-HTT(150Q) mouse model was generated at the time that showed the occurrence of a primary "demyelination" in Huntington disease. Now the authors provide compelling evidence that a specific phosphorylated form of MYRF (at Ser259) binds to mutant HTT. Laquinimod is a polyaromatic compound that had shown significant success in the treatment of demyelination in both human multiple sclerosis and Huntington disease, but its mechanism of action is poorly understood. Here, the authors discover that Laquinimod interferes with the phosphorylation of HTT. In consequence, Laquinimod treatment renders MYRF less likely to be bound by mutant HTT, which in turn improves myelin gene expression and the motor phenotype of PLP-HTT(150Q) mice, but not the early lethal outcome. Expectedly, also knocking-down the identified responsible kinase (PRKG2) increased myelin gene expression in PLP-HTT mice. The authors can confirm their model by showing direct protein interactions.

This is a very interesting set of observation, a comprehensive analysis and first mechanistic insight into HTT-MRF interactions. As the authors discuss, there are likely additional cytotoxic effects of mutant HTT, as the rescue of oligodendroglial dysfunction in PLP-HTT mice is incomplete. In turn, there must be HTT-independent effects of this drug, because the successful treatment of demyelination in MS patients is in the absence mutant HTT expression. Nevertheless, this paper marks the first molecular insight into the obscure role of Laquinimod as a drug that works in human patients. There is certainly room for improvement of some of the figures, including the summary diagram, but I have no major problems with this paper as is. My only concern is the diagnosis of the mutant PLP-HTT mutant phenotype as "demyelination", without showing the activity of macrophages and low grade inflammation. More likely the thinly myelinated axons reflect a developmental problem (dysmyelination), which is nevertheless interesting.

We thank the reviewer for the above suggestions and have incorporated them in the discussion of the revised text. We have revised figures by including high resolution images. We agree with the reviewer that HD affects different types of cells. In our studies, we used HD mouse models that selectively express mutant huntingtin in oligodendrocytes. Thus, the observed effects of LAQ on MYRF and MBP are more likely to be selective to oligodendrocytes. We also followed the reviewer's suggestion to revise the summary diagram, which would be clearer now to show the difference between HD oligodendrocytes and LAQ-treated HD oligodendrocytes.

Referee #3:

The Ms by Yin and colleagues investigate the mechanisms by which Laquinimod (LAQ), a molecule therapeutic interest in MS and HD restores myelination. Indeed, although several studies have described the protective effect of this drug, the molecular mechanism remain elusive. Here,

Yin and colleagues, using a mouse model of HD that selectively express mutant huntingtin (mHTT) in oligodendrocytes and that show demyelination, show that LAQ reduce the level of phosphorylation of the myelin regulatory factor. Authors identified the site of phosphorylation as being S259 by performing MS on immunopurified MYRF. In HD mice, MYRF phosphorylation is increased thus leading to the interaction between MYRF and HTT and subsequent reduction of MBP transcription. Reducing S259 phosphorylation reduces MYRF-HTT interaction and restores transcriptional activity at the MBP promoter. Authors then raised a S259 P specific antibody and found that MYRF phosphorylation is increased in vivo and in HD patients and that phosphorylation is reverted in vivo after LAQ treatment in mice.

How LAQ leads to MYRF S250 phosphorylation? By performing RNA seq on mice treated by LAQ, authors identify PRKG2, a cGMP-activated protein kinase subunit II as being downregulated. By searching for the potential transcription factors that can bind to the PRKG2, authors identify binding sites of Aryl hydrocarbon receptor on PRGK2 promoter and using ChIP assay, found that LAQ acts on Aryl hydrocarbon receptor by increasing its association with the PRGK2 promoter. Next they show that PRGK2 effectively phosphorylate S259 of MYRF both in vitro in cells and in vivo in mouse brain. Finally, authors used CRISPR/Cas9 approach to knock-out PRGK2 in the corpus callosum of PLP-150Q HD mice and observed a reduction of MYRF phosphorylation and increase in MBP expression.

Together, the authors describe a molecular mechanism by which LAQ acts to increase myelination. These findings with the identification of PRGK2 as a mediator of LAQ therapeutic effect is of interest not only for HD but for other disorders with demyelination.

This is a very complete study that should be published as it is given the quality of the experiments and the importance of the findings.

We thank the reviewer for his/her positive comments.

References:

Bujalka H, Koening M, Jackson S, Perreau VM, Pope B, Hay CM, Mitew S, Hill AF, Lu QR, Wegner M, Srinivasan R, Svaren J, Willingham M, Barres BA, Emery B. MYRF is a membrane-associated transcription factor that autoproteolytically cleaves to directly activate myelin genes. *PLoS Biol.* 2013;11(8):e1001625. doi: 10.1371/journal.pbio.1001625. Epub 2013 Aug 13.

Emery B, Agalliu D, Cahoy JD, Watkins TA, Dugas JC, Mulinyawe SB, Ibrahim A, Ligon KL, Rowitch DH, Barres BA Myelin gene regulatory factor is a critical transcriptional regulator required for CNS myelination. *Cell* 138: 172-85 (2009)

Hornig J, Fröb F, Vogl MR, Hermans-Borgmeyer I, Tamm ER, Wegner M. The transcription factors Sox10 and Myrf define an essential regulatory network module in differentiating oligodendrocytes. *PLoS Genet.* 2013 Oct;9(10):e1003907. doi: 10.1371/journal.pgen.1003907. Epub 2013 Oct 31.

Huang B, Wei W, Wang G, Gaertig MA, Feng Y, Wang W, Li XJ, Li S. Mutant huntingtin downregulates myelin regulatory factor-mediated myelin gene expression and affects mature oligodendrocytes. *Neuron.* 2015 Mar 18;85(6):1212-26. doi: 10.1016/j.neuron.2015.02.026.

Meng J, Ma X, Tao H, Jin X, Witvliet D, Mitchell J, Zhu M, Dong MQ, Zhen M, Jin Y, Qi YB. Myrf ER-Bound Transcription Factors Drive C. elegans Synaptic Plasticity via Cleavage-Dependent Nuclear Translocation. *Dev Cell.* 2017 Apr 24;41(2):180-194.e7. doi: 10.1016/j.devcel.2017.03.022.

2nd Editorial Decision

10 March 2020

Thank you for the submission of your revised manuscript. We have now received the enclosed report from referee 1 who still has minor suggestions that I would like you to incorporate before we can proceed with the official acceptance of your manuscript.

A few other changes will also be required:

Su Yang is missing from the author contributions, please add.

I would like to suggest a few changes to the abstract that needs to be written in present tense. Please let me know if you agree with the following:

Demyelination is a common pathological feature of a large number of neurodegenerative diseases including multiple sclerosis and Huntington's disease (HD). Laquinimod (LAQ) was shown to have therapeutic effects on multiple sclerosis and HD. However, the mechanism underlying LAQ function remains unknown. Using HD model mice that selectively express mutant huntingtin in oligodendrocytes and show demyelination, we show that LAQ reduces Ser259 phosphorylation of myelin regulatory factor (MYRF), an oligodendrocyte-specific transcription factor promoting [OK?] the expression of myelin-associated genes. Reduced MYRF phosphorylation inhibits MYRF binding to mutant huntingtin and increases the expression of myelin-associated genes. We also show that PRKG2, the cGMP-activated protein kinase subunit II, promotes Ser259-MYRF phosphorylation and that knocking-down PRKG2 increases myelin-associated protein expression in HD mice. Our findings suggest that PRKG2-mediated phosphorylation of MYRF facilitates demyelination and can serve as a potential therapeutic target to promote remyelination.

EMBO press papers are accompanied online by A) a short (1-2 sentences) summary of the findings and their significance, B) 2-3 bullet points highlighting key results and C) a synopsis image that is 550x200-400 pixels large (the height is variable). You can either show a model or key data in the synopsis image. Please note that text needs to be readable at the final size. Please send us this information along with the revised manuscript.

REFEREE REPORT

Referee #3:

Authors have improved the Ms by modifying text, and replacing images (e.g.: Fig 3D, fig 6G & 7C...)

Reviewer 1 had a problem with the lack of specificity of the effect mediated by Laquinimod. However, the authors use a model in which mHTT is expressed specifically in the oligodendrocytes. Also, authors investigated the effect of LAQ on MBP and MYRF that are specifically expressed in oligodendrocytes.

Reviewer 1 points out the fact that some biochemistry experiments is done in HEK cells or neuroblastomas instead of oligodendrocytes. In addition to the fact that these type of experiments (interactions experiments...) are difficult to perform in primary cultures of oligos, authors have demonstrated their findings in vivo (Fig 6). In particular, the effect of KO of PRKG2 was done in vivo using Crispr/Cas9 approach (Fig 7).

Point 4 of reviewer 1: it might be nice indeed to have EM of myelinated axons after injection of the PRGK2 guided RNA. However, I believe this is not necessary as authors show reduced phosphorylation of MYRF in vivo both by WB and IHC and more importantly an increase of MBP. The limitation of mice with all the animals used for these experiments make that EM studies would require a new cohort of mice which, according to the amount of data already provided may not modify th eoverall conclusion of the study.

For all these reasons, I think the study should be accepted as is.

Some Typos :

Figure 1C legend page 29 : One-way ANNOVA followed with Tukey's test. MBP:

***P=0.0009; MOBP: **P=0.0067; MOG: ***P=0.0003; Data are presented as mean {plus minus} SEM.

2nd Revision - authors' response

10 March 2020

The authors performed all minor editorial changes.

Corresponding Author Name: Shihua Li

Manuscript Number: EMBOR-2019-49783V2